# TRIB3-EGFR interaction promotes lung cancer progression and defines a therapeutic target

Jiao-jiao Yu[1,7], Dan-dan Zhou[1,7], Xiao-xiao Yang[2], Bing Cui [1], Feng-wei Tan[3], Junjian Wang[4], Ke Li[5], Shuang Shang[1], Cheng Zhang[1], Xiao-xi Lv[1], Xiao-wei Zhang [1], Shan-shan Liu[1], Jin-mei Yu [1], Feng Wang[1], Bo Huang[6], Fang Hua [1✉] & Zhuo-Wei Hu[1✉]

High expression or aberrant activation of epidermal growth factor receptor (EGFR) is related to tumor progression and therapy resistance across cancer types, including non-small cell lung cancer (NSCLC). EGFR tyrosine kinase inhibitors (TKIs) are first-line therapy for NSCLC. However, patients eventually deteriorate after inevitable acquisition of EGFR TKI-resistant mutations, highlighting the need for therapeutics with alternative mechanisms of action. Here, we report that the elevated tribbles pseudokinase 3 (TRIB3) is positively associated with EGFR stability and NSCLC progression. TRIB3 interacts with EGFR and recruits PKCα to induce a Thr654 phosphorylation and WWP1-induced Lys689 ubiquitinion in the EGFR juxtamembrane region, which enhances EGFR recycling, stability, downstream activity, and NSCLC stemness. Disturbing the TRIB3-EGFR interaction with a stapled peptide attenuates NSCLC progression by accelerating EGFR degradation and sensitizes NSCLC cells to chemotherapeutic agents. These findings indicate that targeting EGFR degradation is a previously unappreciated therapeutic option in EGFR-related NSCLC.

[1] State Key Laboratory of Bioactive Substance and Function of Natural Medicines, Institute of Materia Medica, Chinese Academy of Medical Sciences & Peking Union Medical College, Beijing 100050, PR China. [2] Department of Medicinal Synthesis Chemistry, Institute of Materia Medica, Chinese Academy of Medical Sciences & Peking Union Medical College, 100050 Beijing, PR China. [3] Department of Thoracic Surgery, National Cancer Center/National Clinical Research Center for Cancer/Cancer Hospital, Chinese Academy of Medical Sciences & Peking Union Medical College, 100021 Beijing, PR China. [4] School of Pharmaceutical Sciences, Sun Yat-sen University, Guangzhou 510006, PR China. [5] Institute of Medicinal Biotechnology, Chinese Academy of Medical Sciences & Peking Union Medical College, 100050 Beijing, PR China. [6] Institute of Basic Medicine, Chinese Academy of Medical Sciences & Peking Union Medical College, 100005 Beijing, PR China. [7] These authors contributed equally: Jiao-jiao Yu, Dan-dan Zhou. ✉email: huafang@imm.ac.cn; huzhuowei@imm.ac.cn

Epidermal growth factor receptor (EGFR) is critical for controlling the growth and survival of epithelial cells and is often targeted therapeutically in epithelial malignancies, including non-small cell lung cancer (NSCLC). Currently, all available EGFR-targeted therapeutics, including small-molecule tyrosine kinase inhibitors (TKIs) and EGFR-targeted monoclonal antibodies, focus on the inhibition of EGFR kinase activity or induction the antibody- and complement-mediated cytotoxicity[1]. EGFR-targeted antibodies are mainly used in the treatment of advanced colorectal and head and neck cancers but not in NSCLC because of their marginal clinical benefit[2]. So far, three generations of EGFR TKIs have been developed to reversibly (the first generation) or irreversibly (the second- and the third generation) inhibit EGFR tyrosine kinase activity and are widely used in NSCLC treatment.

Gefitinib and erlotinib are the first generation of EGFR TKIs that were designed against the wild-type EGFR (WT-EGFR) but show potent and selective inhibitory effect against active EGFR mutations (e.g. exon 19 deletions and the L858R mutation). Second-generation EGFR TKIs (e.g., afatinib and dacomitinib) are irreversible pan-HER blockers that were developed to overcome acquired T790M resistance to first-generation EGFR TKIs but failed because of inacceptable toxicity[3]. The newly developed third-generation EGFR TKIs (e.g., AZD9291 and rociletinib) bind irreversibly to the cysteine-797 residue in the ATP binding site of EGFR, harboring preferential activity for EGFR forms with activating mutations or the T790M resistance mutation relative to the WT-EGFR[4,5]. AZD9291 acts as a dual EGFR/HER2 inhibitor, which moderately decreases aberrant activation of HER2, another mechanism for acquired resistance to EGFR TKIs[6]. Despite success with clinical use, acquired resistance (e.g., the C797S mutation) has already been reported in patients after AZD9291 initiation[7]. Except for the inevitable acquired resistance, EGFR TKIs are more relevant to EGFR-activating mutations than to WT-EGFR, suggesting that this strategy will achieve relatively little benefit in the majority of lung cancer patients who harboring with WT-EGFR. Accumulating evidence has demonstrated that WT-EGFR is critical in the pathogenesis and progression of lung cancer. Elevated WT-EGFR expression not only correlates with acquired resistance to third-generation EGFR TKIs but also participates in the maintenance of mutated KRAS activity and KRAS-driven NSCLC tumorigenesis[8–11]. Thus, there is an urgent need to identify targeted therapeutics against either mutant EGFR or WT-EGFR with alternative mechanisms of action[12].

A number of studies show that EGFR spatial distribution and stability are also crucial determinants in the regulation of lung cancer progression. Even in mutant EGFR-driven lung adenocarcinoma, dysregulation of EGFR degradation further accelerates tumor initiation and progression[13]. Spatial deregulation of EGFR increases the availability of plasma membrane receptors and induces a persistent signaling output[14]. Depletion of sterol-C4-methyl oxidase–like and NAD(P)H steroid dehydrogenase-like protein, two proteins involved in the sterol biosynthesis pathway, inhibits EGFR recycling and sensitizes A431 xenografts to cetuximab treatment[15]. Golgi Membrane Protein 1 interacts with EGFR to promote EGFR recycling to the membrane, leading to prolonged EGFR activation and hepatocellular carcinoma progression[16]. Notably, crosstalk between oncogenic mutations in receptor tyrosine kinases (RTKs) and aberrant RTKs trafficking have been causally linked with human malignancies[17,18]. These findings emphasize that promoting EGFR degradation is an alternative strategy to target EGFR-related cancers.

The pseudokinase Tribble 3 (TRIB3), which acts as a stress sensor in response to a diverse range of stressors, is involved in chronic inflammatory, metabolic, and malignant diseases by interacting with signaling and functional proteins[19]. We recently reported that TRIB3 promoted the initiation and progression of several cancers by interacting with the autophagic receptor p62, impairing the degradation functions of autophagy and proteasome, two critical protein quality control systems in cancer cells. Depletion of TRIB3 results in drastically decreased expression of several tumor-promoting factors, including EGFR, across cancers[20,21]. The Cancer Genome Atlas (TCGA) data show that there exist 1.14% of gene amplification and 1.84% gene mutation of *TRIB3* in NSCLC.

In this study, we identified that the elevated TRIB3 expression is associated with the increases in EGFR stability, recycling, signal activity, and NSCLC progression. We thus assumed that TRIB3 promotes NSCLC through the regulation of EGFR turnover. We found that TRIB3-EGFR interaction results in a series of post-translational modifications of EGFR and thereby enhances the EGFR membrane recycling and signaling activity to support NSCLC stemness. Also, our study reveals the potential utility of disturbing the TRIB3–EGFR interaction in the treatment of NSCLC by accelerating EGFR degradation.

## Results

**TRIB3 is correlated with EGFR and poor survival of NSCLC.** To determine the relationship between TRIB3 and EGFR levels in lung cancer, we detected the expression of these two proteins in several human lung cancer cell lines. High TRIB3 expression was correlated with the elevated EGFR expression in most of the human NSCLC cell lines (Fig. 1a). *TRIB3* depletion not only decreased EGFR expression in these cell lines and in primary NSCLC cells (Fig. 1b), but also suppressed the EGFR-responsive genes in A549 cells (Fig. 1c). We interrogated the TCGA database using online kmplot tools to evaluate 1416 NSCLC patients[22], and identified that high *TRIB3* mRNA level is only correlated with poor survival of lung adenocarcinoma (Supplementary Fig. 1a) but not that of lung squamous carcinoma (Supplementary Fig. 1b). However, high TRIB3 protein was found to be positively correlated with poor survival of both lung adenocarcinoma (Supplementary Fig. 1c, d) and squamous carcinoma (reported in our previous paper, ref. [20].). Consistent with TRIB3 protein expression, higher EGFR protein level was observed in human NSCLC tissue samples than that in the adjacent nontumor tissue samples (Fig. 1d, e). A positive correlation could be observed between TRIB3 and EGFR protein levels in NSCLC tissues (Fig. 1f). Notably, 26% of 147 patients with higher expression of both EGFR and TRIB3 showed significant lower survival rate than patients with single or simultaneous low expression of EGFR and TRIB3 (Fig. 1g).

**TRIB3 enhances EGFR stability and signaling activity.** Because neither the correlation between the mRNA levels of *EGFR* and *TRIB3* from TCGA lung cancer data sets (Supplementary Fig. 1e) nor an effect of *TRIB3* depletion on *EGFR* transcription in A549 cells was detected (Supplementary Fig. 1f), differences in EGFR protein stability were compared between NCI-H157 and A549 cells that showed identical levels of WT-EGFR, but the NCI-H157 cells expressed much less TRIB3 than the A549 cells (Fig. 1a). The half-life of EGFR degradation was over 24 h in the A549 cells but only 3.7 h in the NCI-H157 cells (Supplementary Fig. 1g). Depletion of *TRIB3* in A549 (harboring WT-EGFR) or NCI-H1975 (harboring L858R/T790M double mutations) cells reduced the half-life of EGFR (Fig. 2a and Supplementary Fig. 1h), while overexpression of *TRIB3* in NCI-H157 (harboring WT-EGFR) or NCI-H1650 (harboring mutated EGFR with exon 19 deletion) cells prolonged the half-life of EGFR (Supplementary Fig. 1i, j). These data suggest that TRIB3 positively regulates the stability of

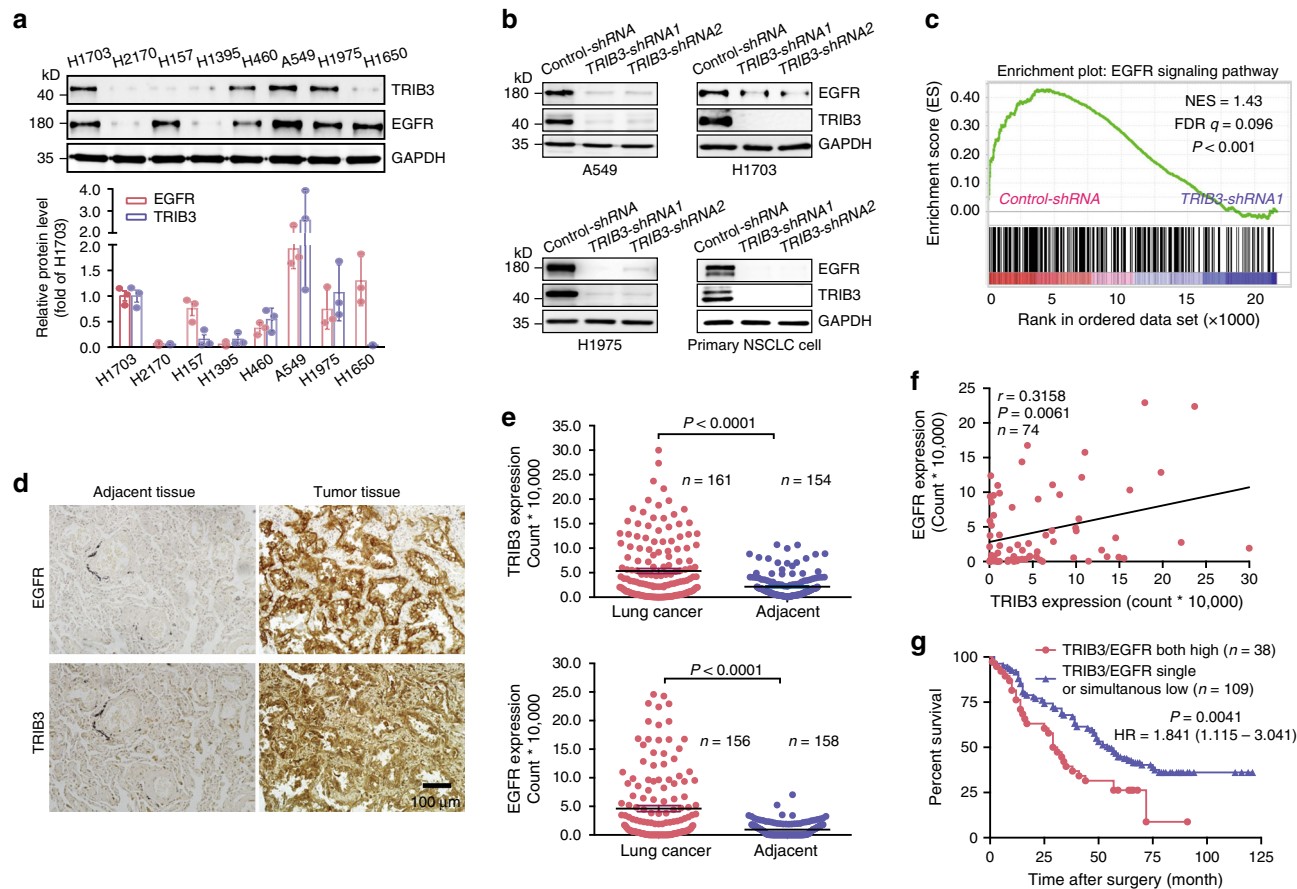

**Fig. 1 TRIB3 expression positively correlates with EGFR in NSCLC. a** Immune-blotting (IB) analyses of TRIB3 and EGFR expression in the indicated NSCLC cell lines. The western blots were quantified by densitometry and calculated relative to GAPDH. The data were normalized as fold of H1703 group and presented as means ± SEMs of three independent biological studies. **b** IB analyses of TRIB3 and EGFR expression in the indicated NSCLC cells stably expressed *control-shRNA* or *TRIB3-shRNAs*. Data are representative immunoblots of three independent assays. **c** GSEA analysis of the "EGFR signaling" gene modules in A549 cells stably expressed *control-shRNA* or *TRIB3-shRNA1*. FDR *q* false discovery rate *q* value, NES normalized enrichment score. **d** Representatives of immunohistochemical staining of TRIB3 (*n* = 161 of lung cancer tissues; *n* = 154 of adjacent tissues) or EGFR (*n* = 156 of lung cancer tissues; *n* = 158 of adjacent tissues). **e** Quantitative analysis of EGFR and TRIB3 expression levels in paired clinical samples. Data are presented as means ± SEM. Statistical significance was determined by two-tailed Student's *t* test. **f** Correlation between TRIB3 and EGFR expression in lung cancer patients at T2 or higher TNM stage. Each point represents the value from one patient. The *P* value is measured by Pearson's rank correlation test. **g** Kaplan–Meier plot of overall survival of patients with lung cancer stratified by TRIB3 and EGFR coexpression level. Patients were divided into two groups: high TRIB3–EGFR expressions vs. single or simultaneous low TRIB3–EGFR expression. Statistical difference was determined by two-sided log-rank test. Source data are provided as a Source Data file.

EGFR, either WT-EGFR or its activating and resistant mutants. Consistently, silencing *TRIB3* protected against EGF-induced phosphorylation of ERK1/2, STAT3/5, while enforced *TRIB3* expression enhanced the phosphorylation of these proteins (Fig. 2b and Supplementary Fig. 1k, l). These data suggest that the elevated TRIB3 expression positively correlates with EGFR expression and its downstream activities by supporting EGFR stability in NSCLCs.

The turnover of EGFR is regulated by endocytosis and postendocytic sorting. Upon stimulation, EGFR is internalized into early endosomes, which are either recycled to the cell surface or undergo lysosomal degradation[23]. In either control or *TRIB3*-depleted A549 cells, EGF stimulation for 30 min induced EGFR internalization and colocalization with early endosome antigen 1 (EEA-1), an early endosome marker (Fig. 2c). However, when EGF stimulation was prolonged to 60 min, EGFR was identified on the cell surface of control A549 cells but not on that of *TRIB3*-depleted A549 cells (Fig. 2c). Similarly, EGF stimulation for 30 min caused EGFR internalization and colocalization with EEA-1 in control or *TRIB3*-overexpressing NCI-H157 cells; however,

membrane EGFR expression was observed only in *TRIB3*-overexpressing NCI-H157 cells stimulated with EGF for 60 min (Supplementary Fig. 2a). These data indicate that elevated TRIB3 expression enhances membrane EGFR expression without impeding EGFR endocytosis. Thus, we suspected that TRIB3 upregulated EGFR expression by enhancing EGFR recycling. Stimulation of cells with EGF for 60 min did not reduce the membrane EGFR level in control A549 cells but did reduce the level in control A549 cells pretreated with monensin, a recycling inhibitor, indicating that EGFR is recycled back to the cell surface after 60 min of EGF treatment. However, the membrane EGFR level in *TRIB3*-depleted A549 cells, which were not pretreated with monensin, reduced with 60 min of EGF treatment (Fig. 2d and Supplementary Fig. 2b). Consistently, EGF stimulation reduced the membrane EGFR level in control NCI-H157 cells but not in NCI-H157 cells overexpressing *TRIB3*; additionally, monensin reduced the EGFR membrane level in *TRIB3*-over-expressing NCI-H157 cells (Supplementary Fig. 2b, c). These data indicate that TRIB3 enhances membrane EGFR expression by promoting EGFR recycling. Indeed, after 120 min of EGF

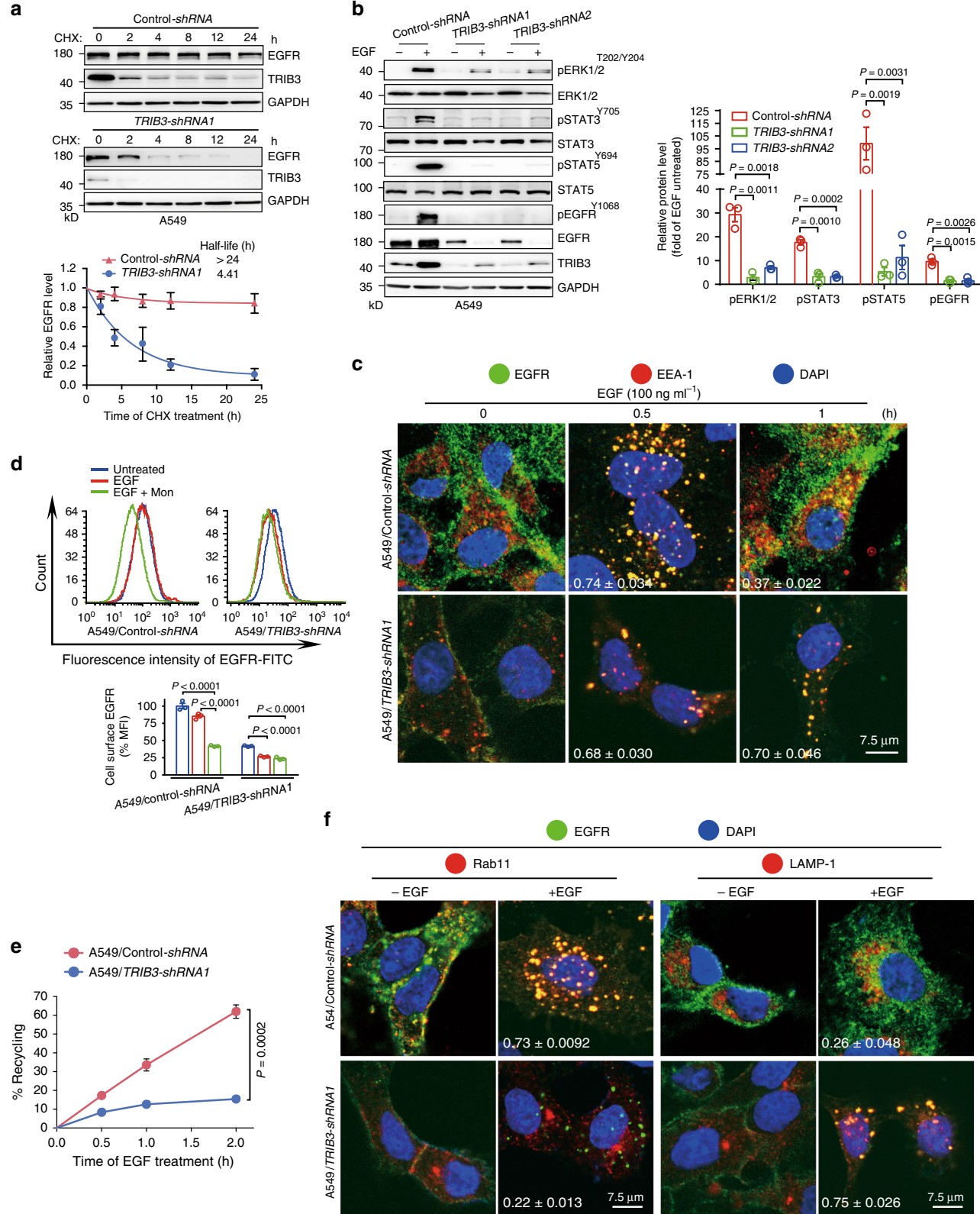

stimulation, the recycling rate of EGFR was >60% in control A549 cells, but it was only ~10% in *TRIB3*-depleted A549 cells (Fig. 2e). Likewise, overexpression of *TRIB3* in NCI-H157 and NCI-H1650 cells increased the recycling rate of EGFR after EGF stimulation (Supplementary Fig. 2d, e). Moreover, a substantial amount of EGFR was predominantly colocalized with Rab11-positive

recycling vesicles in control A549 cells or in *TRIB3*-overexpressing NCI-H157 cells after EGF stimulation; however, far less EGFR was colocalized with lysosomal associated membrane protein 1 (LAMP-1), a lysosomal degradation marker (Fig. 2f and Supplementary Fig. 2f). These results suggest that the elevated TRIB3 expression supports EGFR stability and enhances EGFR recycling.

**Fig. 2 TRIB3 promotes EGFR recycling to enhance its stability and signaling activity. a** Control or *TRIB3*-silenced A549 cells were treated with cycloheximide (CHX) (10 µg ml$^{-1}$) at indicated intervals and protein stability of EGFR was analyzed by IB. Data are means ± SEM of four independent assays. **b** A549 cells stably expressed *control-shRNA* or *TRIB3-shRNA* plasmid were stimulated with or without EGF (100 ng ml$^{-1}$) for 1 h. Indicated proteins were analyzed by IB analysis. Data represent means ± SEM of three independent assays. **c** Confocal microscopy images show the distribution of EGFR and early endosome antigen 1 (EEA-1) in control or *TRIB3*-silenced A549 cells before and after 0.5 or 1 h of EGF (100 ng ml$^{-1}$) stimulation. Quantification of EGFR and EEA-1 colocalization was shown as Pearson's coefficient. Data are means ± SEM of three independent assays. **d** Quantitative analyses of cell surface EGFR in A549 cells stably expressed *control-shRNA* or *TRIB3-shRNA1*. Cells were preincubated with DMSO or 10 µM monensin for 4 h, and then treated with or without EGF (100 ng ml$^{-1}$) for another 1 h. The Mean Fluorescence Intensity (MFI) of EGFR on cell surface was detected by flow cytometry analysis. Top: representative flow cytometry data. Bottom: the data were normalized to A549/*control-shRNA* cells without EGF stimulation, which was considered as 100% MFI signal. Data are means ± SEM of three independent assays. **e** EGFR recycling was detected in A549 cells stably expressed *control-shRNA* or *TRIB3-shRNA1* plasmid. Data are means ± SEM of three independent assays. **f** Colocalization analysis of EGFR with Rab11 and LAMP-1 before or after 30 min EGF treatment in control and *TRIB3*-silenced A549 cells. Quantification of EGFR/Rab11 or EGFR/LAMP-1 colocalization was shown as Pearson's coefficient. Data represent means ± SEM of three independent assays. Statistical significance between two groups was determined with two-tailed Student's *t* test. Statistical significance among groups was determined by one-way ANOVA test. Source data are provided as a Source Data file.

**TRIB3 promotes PKCα-mediated EGFR Thr654 phosphorylation.** TRIB3 induces multiple cellular functions through protein–protein interactions[19]. To determine the critical mediator of TRIB3-enhanced EGFR recycling, a high-throughput protein array screening was carried out, and protein kinase C alpha (PKCα) was identified as a binding partner of TRIB3 (Fig. 3a). Notably, co-immunoprecipitation (CO-IP) assays showed that TRIB3, EGFR, and PKCα were coprecipitated by each antibody (Fig. 3b). In addition, TRIB3, EGFR, and PKCα were colocalized in the cytoplasm upon EGF stimulation (Fig. 3c). Thr654 in EGFR, a major phosphorylation site target by PKCα, has been reported to divert the internalized EGFR from a degradative pathway into the recycling endosomes[24]. Under EGF treatment, a substantial amount of EGFR (77 ± 4.3%) was colocalized with PKCα in control A549 cells (Fig. 3d, left); in TRIB3-depleted cells, the colocalization was much less (12 ± 2%) than that in the control cells, and accompanied by reduced total amounts of EGFR and PKCα (Fig. 3d, right). TRIB3 depletion reduced EGF-induced EGFR$^{T654}$ phosphorylation even when PKCα was ectopically expressed (Fig. 3e). These data suggest that TRIB3 acts as a scaffolding to assemble a heterotrimeric complex for PKCα-induced EGFR$^{T654}$ phosphorylation and subsequent membrane recycling.

To map the interaction region of TRIB3 that interacts with EGFR and PKCα, deletion mutants of HA-tagged TRIB3 were constructed and subjected to CO-IP assay. The C-terminus of the TRIB3 kinase-dead (KD) region (M4 mutant) was identified to interact with EGFR (Fig. 3f). However, it was the C-terminal tail of TRIB3 that was responsible for the association between TRIB3 and PKCα (Fig. 3g). Furthermore, the intracellular juxtamembrane (JM) region of EGFR was identified to interact with TRIB3 (Fig. 3h, i). Moreover, restoring the expression of TRIB3 but not the *KDC* deletion mutant (M5) reversed the inhibitory effect of TRIB3 depletion on EGFR recycling (Fig. 3j). These data indicate that the interaction between TRIB3 and EGFR is crucial for EGFR recycling.

**Lys689 ubiquitination is a decisive signal for EGFR recycling.** Phosphorylation and ubiquitination are two key modifications for the ligand-induced endocytosis and degradation of EGFR[25]. We evaluated the effect of TRIB3 on EGFR ubiquitination. Overexpression of TRIB3 enhanced Lys63 (K63)- but not K48-linked EGFR ubiquitination (Supplementary Fig. 3a), and the modification occurred in the JM region of EGFR (Supplementary Fig. 3b). These findings were verified by the abrogation of TRIB3-enhanced JM ubiquitination by the Lys63-to-Arg (K63R) ubiquitin mutant (Fig. 4a). There are five Lys sites (K652, K684, K689, K690, and K692) in the JM region of EGFR. We found that K689R mutation accelerated EGFR degradation, but other

mutations did not (Fig. 4b). Using JM-4KR, a mutant harboring 4 Lys-to-Arg mutations (not K689), TRIB3-enhanced K63-linked ubiquitylation was found to occur at K689 (Fig. 4c). Notably, a JM-4KR mutant with a phosphorylation-resistant Thr654-to-Ala mutation (referred to as JM-4KR/T654A) lost the ability to be ubiquitinated upon *TRIB3* overexpression (Fig. 4c), suggesting that EGFR T654 phosphorylation is critical for the K63-linked K689 ubiquitination. Moreover, the K689R mutation reduced the recycling rate of the phosphomimetic Thr654-to-Asp (T654D) EGFR mutant, suggesting that the K63-linked K689 ubiquitination of EGFR is a decisive signal for EGFR recycling (Fig. 4d).

SMAD Specific E3 Ubiquitin Protein Ligase 2 (SMURF2) and Ring Finger Protein 126 (RNF126), two E3 ligases reported to interact with EGFR to promote its ubiquitination and stability[26,27], could not induce K63-linked ubiquitination at K689 in the JM region (Supplementary Fig. 3c, d). To identify EGFR-associated E3 ligases in the context of high TRIB3 expression, IP of lysates from A549 cells with an anti-EGFR antibody was undertaken, followed by mass spectrometry-based proteomic analysis (Supplementary Fig. 3e). Five E3 ligases were identified from the precipitated proteins (Supplementary Table 1). Among them, the C-terminus of Hsc70-interacting protein (CHIP), an EGFR degradation-promoting E3 ligase[28], and RNF126 could be excluded from the above discussion. Of the remaining three E3 ligases, only the WW domain-containing E3 ubiquitin protein ligase 1 (WWP1) enhanced EGFR expression driven by the cytomegalovirus promoter (Supplementary Fig. 3f). WWP1 interacted with EGFR, and the T654A mutation impaired the association between EGFR and WWP1 (Fig. 4e). Indeed, knocking down of either PKCα or WWP1 expression destroyed TRIB3-enhanced EGFR recycling, confirming the critical roles of these two proteins in EGFR recycling (Fig. 4f). WWP1 increased K63-linked ubiquitination of WT-EGFR but not that of the K689R mutant (Supplementary Fig. 3g). However, the T654A mutation protected EGFR from WWP1-induced K689 ubiquitination (Fig. 4g), suggesting that T654 phosphorylation is a necessary signal for WWP1-induced K689 ubiquitination. Overexpression of *WWP1* inhibited the WT-EGFR degradation but not K689R mutant degradation (Supplementary Fig. 3h). However, TRIB3-enhanced EGFR stability was abolished when *WWP1* was depleted (Supplementary Fig. 3i), indicating that TRIB3 enhances EGFR stability via WWP1. These data suggest that TRIB3 interacts with EGFR to recruit PKCα and phosphorylate EGFR at T654, which causes WWP1-catalyzed, K63-linked K689 ubiquitination of EGFR, a decisive signal for EGFR recycling, and enhances EGFR stability.

Because both PKCα and WWP1 are necessary for TRIB3-promoted EGFR recycling, we detected the expression of these two proteins in human NSCLC samples. Higher PKCα and

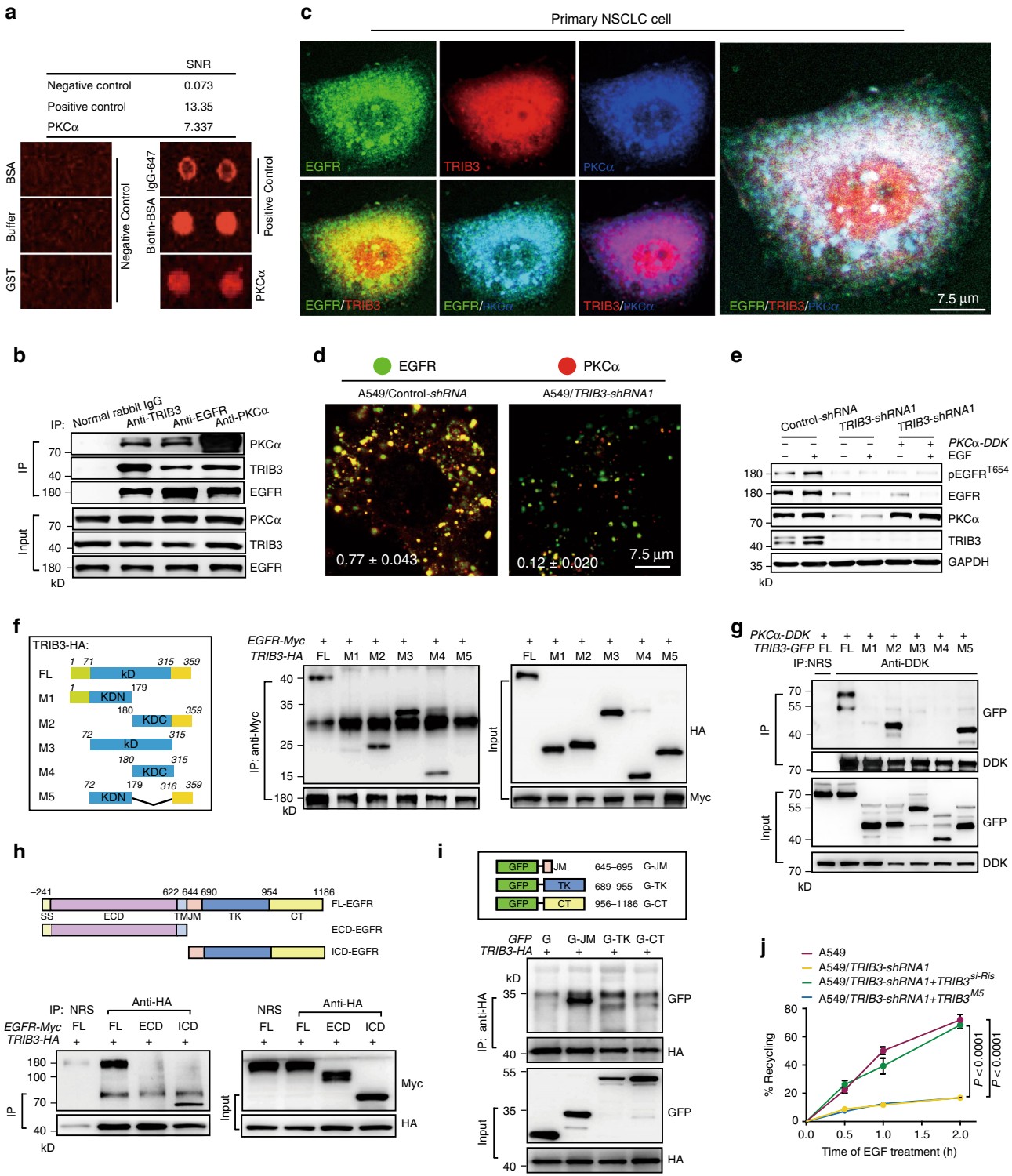

WWP1 expression was observed in human NSCLC tissue samples than in adjacent nontumor tissue samples (Fig. 5a, b and Supplementary Fig. 4a, b). Notably, elevated expression of either PKCα or WWP1 correlated with a poor survival rate in NSCLC patients (Fig. 5c, d). Indeed, depletion of either PKCα or WWP1 abrogated TRIB3-enhanced proliferation, invasion, and tumor growth (Fig. 5e–h). Importantly, these effects could be rescued by overexpressing the EGFR$^{T654D}$ mutant but not the EGFR$^{T654A}$ or EGFR$^{T654D/K689R}$ mutants (Supplementary Fig. 4c, d), indicating that TRIB3 promotes NSCLC progression by enhancing PKCα

and WWP1-regulated EGFR stability. Cancer stem cells (CSCs) are considered as the root of tumor relapse and resistance[29]. TRIB3 depletion decreased the expression of critical CSC markers (Supplementary Fig. 4e), as well as their target gene sets (Supplementary Fig. 4f). Indeed, depletion of either PKCα or WWP1 abrogated TRIB3-enhanced tumor sphere formation (Fig. 5i); while TRIB3 depletion-impaired tumor sphere formation could be rescued by overexpressing the of EGFR$^{T654D}$ mutant but not the EGFR$^{T654A}$ or EGFR$^{T654D/K689R}$ mutants (Supplementary Fig. 4g). These data suggest that TRIB3, PKCα,

**Fig. 3 TRIB3 interacts with EGFR to promote PKCα-mediated EGFR phosphorylation. a** Interactors of TRIB3 were screened through a human protein microarray (HuProt™ 20 K) with purified TRIB3 protein. Data shown are signal to noise ratio (SNR) values and representative images of PKCα. Alexa647-labeled IgG and Biotin-labeled BSA were used as positive controls. GST protein, the buffer only and BSA were used as negative controls. Two repeats were designed for each protein of the microarray. **b**, **c** TRIB3, EGFR and PKCα form heterotrimeric complex. Protein extracts from A549 cells were IP with anti-TRIB3, anti-EGFR or anti-PKCα Abs individually, and detected with indicated Abs (**b**). Primary NSCLC cells were treated with EGF (100 ng ml⁻¹) for 30 min. Colocalization of EGFR with PKCα and TRIB3 was analyzed with confocal microscopy (**c**). **d** Confocal microscopy images show the colocalization of EGFR with PKCα in control and *TRIB3*-silenced A549 cells treated with EGF (100 ng ml⁻¹) for 30 min. Quantification of EGFR and PKCα colocalization was shown as Pearson's coefficient. Data are means ± SEM of three independent assays. **e** Control and *TRIB3*-silenced A549 cells were transfected with indicated plasmid and treated with or without EGF (100 ng ml⁻¹) for 1 h. EGFR T654 phosphorylation was detected by IB. **f** Mapping TRIB3 regions involved in EGFR binding. Left: schematic diagram of TRIB3 deletion mutants. Right: HEK 293T cells were cotransfected with indicated constructs of *EGFR-Myc* and *TRIB3-HA* deletion mutants. Cell extracts were IP with anti-Myc Ab. **g** Mapping TRIB3 regions involved in PKCα binding. **h** Mapping EGFR regions involved in TRIB3 binding. **i** Mapping the domains of ICD region involved in TRIB3 binding. **j** The recycling of EGFR was determined by ELISA in A549 cells with the indicated constructs stably expressed. Data represent mean ± SEM of three assays. Data in **e–i** are representatives of three independent assays. Statistical significance was determined by one-way ANOVA test. (KD kinase-dead region, KDN N-terminal of KD, KDC C-terminal of KD, ECD extracellular domain, ICD intracellular domain. JM juxtamembrane, TK tyrosine kinase, CT C-terminal, TRIB3^si-Ris the *TRIB3-siRNA*-resistant expressing plasmid). Source data are provided as a Source Data file.

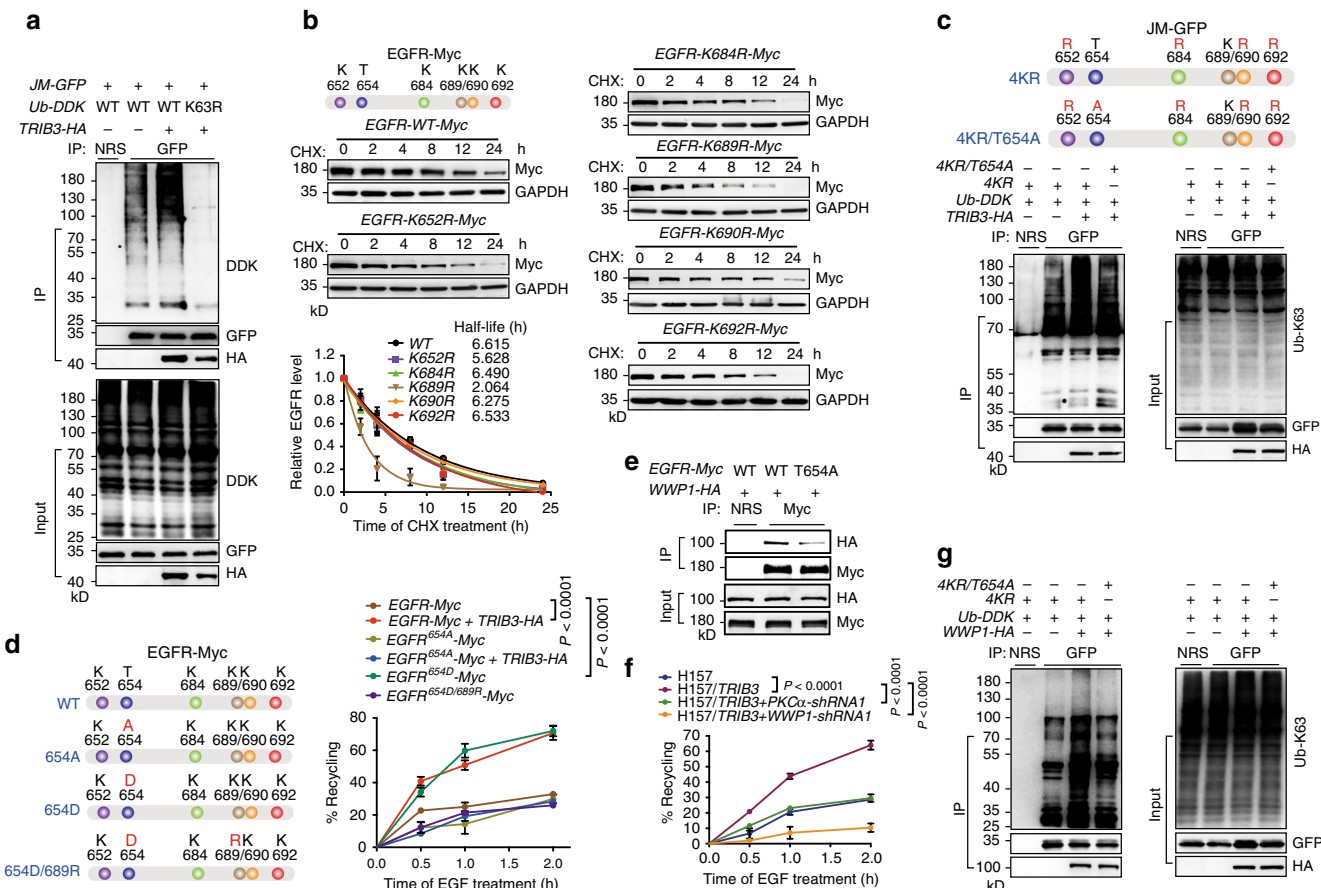

**Fig. 4 WWP1-induced EGFR K689 ubiquitination is a decisive signal for EGFR recycling. a** Cell extracts from HEK 293T cells transfected with indicated plasmids were IP with anti-GFP Ab. The ubiquitination of EGFR JM region was detected by IB. **b** Mapping the lysine sites responsible for EGFR stability in EGFR JM region. HEK 293T cells transfected with indicated plasmids were incubated with CHX (10 μg ml⁻¹) at indicated intervals and expression of indicated proteins were analyzed by IB. **c** The K63-linked ubiquitination of the indicated JM mutants were analyzed by IB in HEK 293T cells transfected with indicated plasmids. **d** The recycling of indicated EGFR mutants was detected in HEK 293T cells transfected with indicated plasmids. **e** Cell extracts from HEK 293T cells transfected with indicated plasmids were IP with anti-Myc Ab. The binding of WWP1 with EGFR wild type and EGFR T654A mutant were detected by IB. **f** EGFR recycling was determined by ELISA in indicated NCI-H157 cells. **g** The K63-linked ubiquitination of the indicated JM mutants were analyzed by IB in HEK 293T cells transfected with indicated plasmids. For **a**, **c**, **e**, and **g**, data are representatives of three independent assays. For **b**, **d**, and **f**, data indicate means ± SEM, n = 3. Statistical significance among groups was determined by one-way ANOVA test. Source data are provided as a Source Data file.

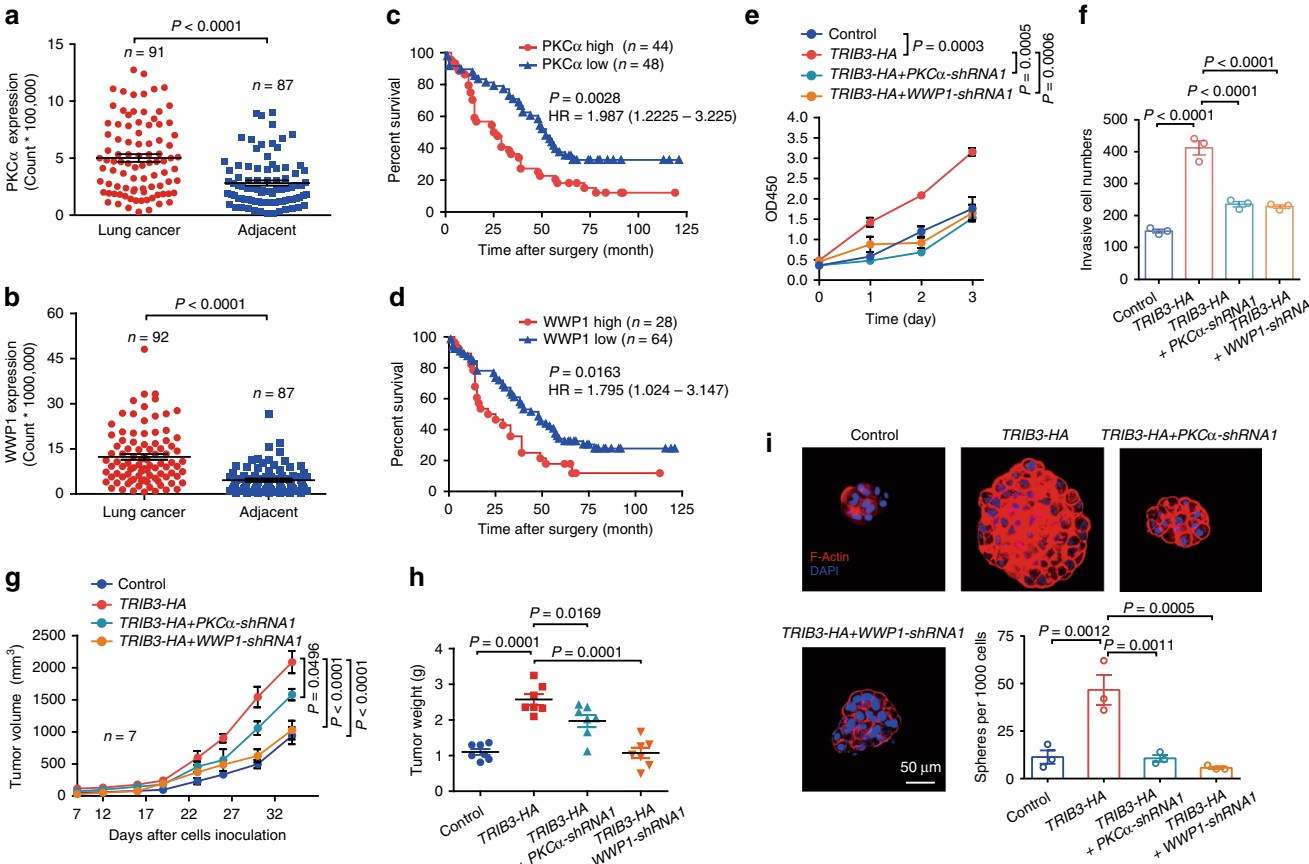

**Fig. 5 PKCα and WWP1 positively correlate with poor survival and NSCLC stemness.** Quantitative analysis of PKCα (**a**) and WWP1 (**b**) expressions in paired clinical samples. Statistical significance was determined by two-tailed Student's $t$ test. Kaplan–Meier plot of overall survival of patients with lung cancer stratified by PKCα (**c**) and WWP1 (**d**) protein expression levels. Statistical difference was determined by two-sided log-rank test. **e** Cell proliferation was measured by CCK-8 assay in NCI-H157 cells stably transfected with indicated plasmids. Data represent means ± SEM of three assays. **f** The invasive capacities of NCI-H157 cells stably transfected with indicated plasmids were evaluated with transwell assays. Data represent means ± SEM of three assays. Tumor growth curves (**g**) and quantified tumor weight (**h**) of mice subcutaneously inoculated with NCI-H157 cells stably transfected with indicated plasmids. Data are presented as means ± SEM, $n = 7$. **i** Immunostaining for F-Actin and DAPI in the tumor sphere of NCI-H157 cells with indicated plasmids stably expressed. Data represent means ± SEM of three assays. Statistical significance between two groups was determined with two-tailed Student's $t$ test. Statistical significance among groups was determined by one-way ANOVA test. Source data are provided as a Source Data file.

WWP1, and EGFR form a regulatory axis to promote tumor stemness and progression in NSCLC by maintaining EGFR stability.

**Disturbing TRIB3–EGFR interaction reduces EGFR stability.** To verify the critical role of TRIB3–EGFR interaction in the regulation of EGFR stability and recycling, two α-helices were identified in the JM region of EGFR by a predictive I-TASSER server based on the PDB data (1Z9I) from an nuclear magnetic resonance analysis of EGFR JM region (Fig. 6a, Left)[30,31]. By fusing the two α-helical peptides with GFP via a flexible linker, JMA2 was found to be the main α-helix responsible for the EGFR–TRIB3 interaction (Fig. 6a, Right). However, the isolated JMA2 peptide showed no binding with TRIB3 (Fig. 6b). As the I-TASSER prediction shows discontinuous α-helix in the JMA2 peptide at Leu683 and Lys684, we tried amino acid substitution with the ones showing similar physicochemical properties. Leu-to-Arg (basic amino acids with aliphatic group) substitution (denoted as JGZ hereinafter) displayed an acceptable binding affinity with TRIB3; while Leu-to-Ile or Leu-to-Val (branched-chain amino acids) did not (Fig. 6b). To assess the contribution of amino acids to JGZ-TRIB3 binding in JGZ peptide, each amino acid residue of JGZ was substituted with Ala. The residues

Leu680, Ile682, Leu683, Arg684, and Lys689 were critical for the binding of JGZ to TRIB3 or for the maintaining of the α-helical conformation, because these mutations abolished JGZ-TRIB3 binding (Supplementary Fig. 5a). To optimize the physicochemical properties of the peptide, we inserted chemical staples at four ($i$, $i + 4$) positions to generate peptides SAH-JGZ1 ~ SAH-JGZ4 (Fig. 6c). The insertion of $i$, $i + 4$ staples enhanced the α-helical content up to fourfold over the content of the unmodified JGZ peptide (Fig. 6d). However, only SAH-JGZ4 showed an improved binding affinity with TRIB3, detecting with either surface plasmon resonance analysis (Fig. 6e and Supplementary Fig. 5b) or fluorescence polarization (FP) binding and competition experiments (Supplementary Fig. 5c, d). The chloroalkane penetration assay was used to quantify the cytosolic delivery of SAH-JGZ4 using A549 cells stably expressed Halo-GFP-Mito, the cytosolically oriented GFP-haloenzyme[32,33] (Fig. 6f). Fluorescence confocal microscopy showed that preincubation with chloroalkane-tagged SAH-JGZ4 (denoted as ct-SAH-JGZ4) suppressed almost of the cytosolic ct-TAMRA signal, while weaker suppressive effect was observed in cells preincubated with ct-Pep2-JGZ, a chimeric peptide fused with the cell-penetrating peptide Pep2[34] (Fig. 6g). Moreover, ct-SAH-JGZ4 showed a dose-dependent suppression of the ct-TAMRA signal with a $CP_{50}$ (concentration at which 50%

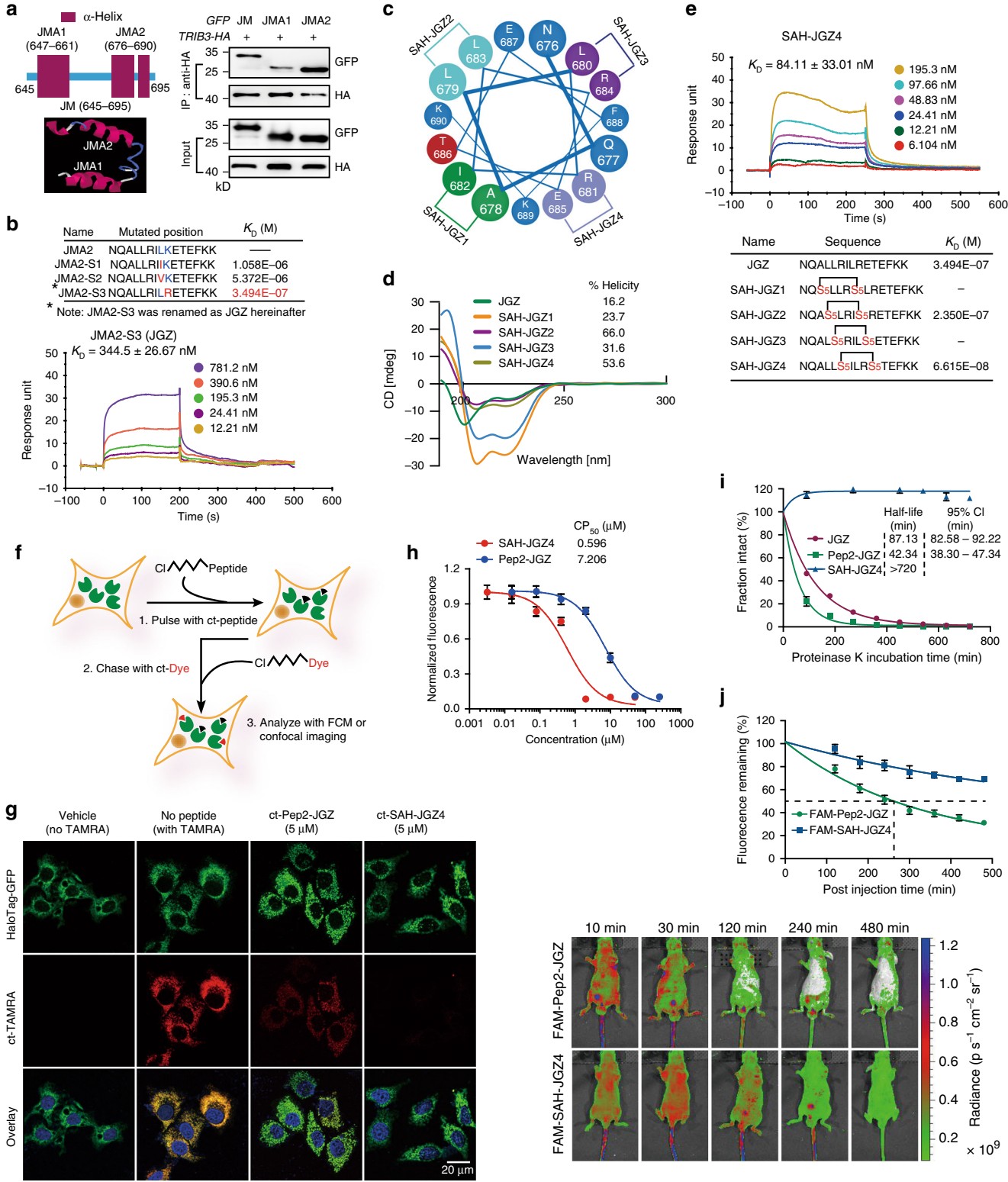

cell penetration was observed) of 0.596 μM, more than tenfold lower than that of the ct-Pep2-JGZ (Fig. 6h). These results confirmed the enhanced cell penetration activity and the cytosolic localization of SAH-JGZ4. To measure the comparative protease susceptibility of unmodified and stapled SAH-JGZ4 peptides, the peptides were subjected to proteinase K or pepsin digestion, followed by liquid chromatography/mass spectrometry analysis of the reaction products over time. Compared with the unmodified

JGZ template, the SAH-JGZ4 peptide conferred a marked enhancement in the half-life, demonstrating the striking proteolytic resistance of the stapled SAH-JGZ4 peptide (Fig. 6i and Supplementary Fig. 5e). Moreover, SAH-JGZ4 not only showed about 100-fold increased plasma stability than that of JGZ (Supplementary Fig. 5f), but also displayed a slower half-clearance rate than Pep2-JGZ (Fig. 6j). Taken together, these data indicate that stapled SAH-JGZ4 not only confers a robust

**Fig. 6 Generation and physiochemical analysis of TRIB3-binding stapled peptide. a** Identification of TRIB3-binding α-helical peptide in EGFR JM region. Left: I-TASSER server analysis of the secondary structure of EGFR JM region. Right: the JM, JMA1, and JMA2 fragments were constructed into *pEGFP-C1* expression vector. The interactions of TRIB3 with the three fragments were evaluated with CO-IP assays. Data are representatives of three assays. **b** Kinetic interactions of JMA2 and its derivative peptides with TRIB3 were determined by surface plasmon resonance analyses ($n = 3$). **c** Structural scheme of the stapled $i, i + 4$–JGZ peptide helices. **d** The helicity of the SAH-JGZs peptides were determined by circular dichroism analysis. **e** Kinetic interactions of stapled peptides with TRIB3 were determined by surface plasmon resonance analyses. Top: the representative SPR diagram of SAH-JGZ4. Bottom: the sequences of stapled peptide and their dissociation constant ($n = 3$). **f** Schematic diagram of chloroalkane penetration assay (CAPA). **g** Representative images of HaloTag stably expressed A549 cells pulsing with 5 μM indicated chloroalkane-tagged peptides for 4 h, and then chasing with HaloTag TAMRA ligand. Data are representatives of three independent assays. **h** HaloTag stably expressed A549 cells were pulsed with different concentrations of chloroalkane-tagged peptides for 4 h, and then chasing with HaloTag TAMRA ligand. The mean TAMRA fluorescence was determined by fluorescence microplate reader. The data were normalized using a nonpulsed control as 100% signal. **i** Proteinase K resistance profiles of JGZ, Pep2-JGZ, and SAH-JGZ4. The reaction productions were evaluated by LC/MS analysis. **j** IVIS Spectrum analyses of the distributions and clearance of FAM-Pep2-JGZ and FAM-SAH-JGZ4 at indicated times. Data in **h–j** are means ± SEM of three independent assays. Source data are provided as a Source Data file. (Asn: N, Gln: Q, Ala: A, Leu: L, Arg: R, Ile: I, Lys: K, Glu: E, Thr: T, Phe: F).

cell-penetrating ability and cytosolic localization, but also enables intracellular protease resistance.

Using this stapled peptide as a probe, we verified the molecular mechanism and critical roles of the TRIB3–EGFR interaction in NSCLC progression. Treatment of A549 or primary human NSCLC cells with SAH-JGZ4 suppressed the interactions of TRIB3–EGFR and EGFR–PKCα (Supplementary Fig. 6a, b). In addition, SAH-JGZ4 treatment inhibited the T654 phosphorylation of EGFR in A549 cells and reduced the expression of EGFR and PKCα (Supplementary Fig. 6c, top). In HEK 293T cells, using ectopic expression to maintain the identical EGFR and PKCα expression levels, we further confirmed that SAH-JGZ4 treatment reduced EGFR T654 phosphorylation by disturbing the TRIB3–EGFR and EGFR–PKCα interactions (Supplementary Fig. 6c, bottom). Furthermore, SAH-JGZ4 inhibited EGFR recycling (Supplementary Fig. 6d) and promoted EGFR degradation (Supplementary Fig. 6e). Also, SAH-JGZ4 inhibited basal and EGF-induced phosphorylation of ERK1/2, STAT3/5, and EGFR (Supplementary Fig. 6f). These data indicate that SAH-JGZ4 suppresses EGFR recycling and downstream signaling activities and promotes EGFR degradation by disturbing the TRIB3–EGFR interaction.

**Targeting EGFR degradation inhibits lung cancer development**. Because SAH-JGZ4 could promote EGFR degradation and suppress EGFR signaling activity, its antitumor effect was evaluated using in vitro and in vivo models. SAH-JGZ4 not only protected against the EGF-induced expression of core pluripotency factors (Supplementary Fig. 6g), but also suppressed tumor proliferation, invasion, and intrinsic oncosphere formation in A549 cells (Supplementary Fig. 6h–j). SAH-JGZ4 treatment suppressed tumor growth in subcutaneous xenograft models with A549 cells in a dose-dependent manner and it showed a better antitumor effect than gefitinib at a dose of 2 mg kg$^{-1}$ administered twice a week (Fig. 7a and Supplementary Fig 7a). SAH-JGZ4 induced a dose-dependent reduction of metastasis to the liver, which was better than the reduction induced by gefitinib (Fig. 7b). Metastasis was confirmed by anti-human mitochondria antibody staining, which is a marker for human cells in a xenograft model (Supplementary Fig. 7b). Notably, SAH-JGZ4 also suppressed tumor growth and metastasis in mice inoculated with NCI-H1975 cells, which are gefitinib-resistant lung cancer cells harboring the T790M mutation (Fig. 7c, d and Supplementary Fig. 7c); the antitumor efficacy of SAH-JGZ4 was superior to that of gefitinib and comparable to that of AZD9291 (Fig. 7c, d and Supplementary Fig. 7c). Mechanistically, SAH-JGZ4 disturbed the in vivo interactions of EGFR–TRIB3 and EGFR–PKCα, and suppressed the expression of EGFR and PKCα in the inoculated tumor tissues (Fig. 7e). In addition, the phosphorylation of

STAT3/5 and EGFR, as well as the expression of total EGFR, PKCα, TRIB3, and core pluripotency factors, were reduced in tumor tissue samples from NCI-H1975 inoculated mice treated with SAH-JGZ4 (Supplementary Fig. 7d, e). Using a lung orthotopic transplantation model, we found that SAH-JGZ4 attenuated the metastasis of A549 cells from the inoculated side to the opposite side (Fig. 7f, g). Furthermore, SAH-JGZ4 treatment increased the survival rate of the tumor-bearing mice orthotopically inoculated with either A549 or NCI-H1975 cells (Fig. 7h, i). SAH-JGZ4 reduced the tumor-initiating cell (TIC) frequency by sixfold (1/TIC from ~93 cells to ~546 cells) (Fig. 7j, k), suggesting that targeting EGFR stability is a potential strategy to inhibit the stemness of lung cancer cells. Indeed, SAH-JGZ4 treatment decreased the size of tumor organoids derived from NSCLC patients (Fig. 7l).

CSCs are considered a main player for chemoresistance against a variety of drugs[35]. Indeed, SAH-JGZ4 treatment enhanced the carboplatin- and pemetrexed-induced cell death in A549 cells, but not that of gemcitabine and taxol (Fig. 8a). The sensitization effects for carboplatin and pemetrexed were also observed in NCI-H1975 cells (Fig. 8b). We next examined whether SAH-JGZ4 could sensitize the therapeutic effect of carboplatin or pemetrexed in two NSCLC patient-derived xenograft (PDX) models, which showed higher TRIB3 and EGFR expression than A549 cells (Fig. 8c and Supplementary Fig. 7f). Combination of SAH-JGZ4 with carboplatin or pemetrexed showed drastic inhibition effects on tumor growth and further prolonged the survival rate of tumor-bearing mice in both of the two NSCLC PDX models (Fig. 8d–k). These data suggest that accelerating EGFR degradation is a potential therapeutic strategy to enhance the sensitivity of carboplatin- or pemetrexed-based chemotherapy in NSCLC. Although long-term AZD9291 treatment reduced TRIB3 and EGFR expression, it upregulated the expression and phosphorylation of STAT3, STAT5, and ERK1/2 (Supplementary Fig. 7d), suggesting the compensatory activation of other oncogenic pathways and induction of cancer stemness. Except for the acquisition of a T790M mutation in EGFR exon 20, the emergence of bypass signaling pathways such as c-MET, HER2, insulin-like growth factor 1 receptor (IGF1R), fibroblast growth factor receptor 1 (FGFR1), and AXL are also critical resistance mechanisms[36–40]. SAH-JGZ4 showed no effect on the phosphorylation and expression of HER2, while AZD9291 displayed suppressive effect on HER2 phosphorylation as reported[6] (Supplementary Fig. 7g). Moreover, SAH-JGZ4 treatment decreased the expression and promoted the degradation of c-MET, but not that of FGFR1, IGF1R, and AXL (Supplementary Fig. 7h, i). These data indicated that SAH-JGZ4 sensitized lung cancer cells to carboplatin or pemetrexed chemotherapy, which may rely on the comprehensive effects of not only promoting

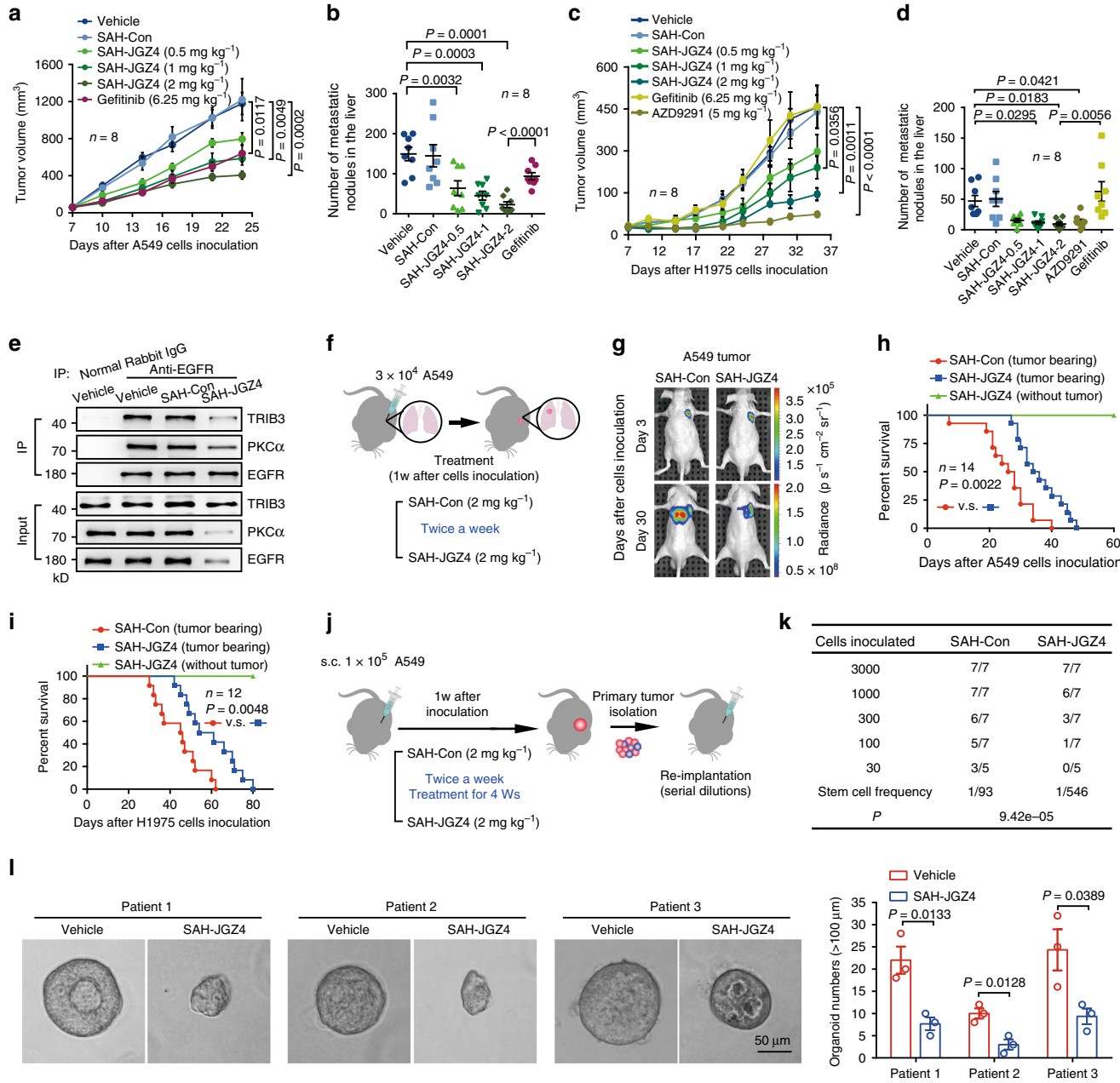

**Fig. 7 Targeting EGFR stability inhibits lung cancer initiation and progression. a** A549 tumor growth curves under different treatment. **b** Quantitative analysis of metastatic nodule numbers from mice as described in **a**. **c** NCI-H1975 tumor growth curves under different treatment. **d** Quantitative analysis of metastatic nodule numbers from mice as described in **c**. Data in **a**–**d** are presented as means ± SEM, $n = 8$. **e** The EGFR–TRIB3 or EGFR–PKCα interaction in xenograft tumors from indicated mice were detected with CO-IP assays. Data are representatives of three assays. **f** Generation of orthotopic xenograft lung cancer model for investigating in vivo antitumor activity of SAH-JGZ4. **g** Representatives of bioluminescence images of orthotopic lung implanted mice at indicated times. **h** Survival curves for mice orthotopically lung implanted with A549 cells under indicated treatment. Statistical difference was determined by two-sided log-rank test, $n = 14$. **i** Survival curves for mice orthotopically lung implanted with NCI-H1975 cells under indicated treatment. Statistical difference was determined by two-sided log-rank test, $n = 12$. **j** Strategy for the evaluation of SAH-JGZ4 on tumor initiation capacity in vivo. **k** Frequency of tumorigenic cell and probability estimates were calculated according to Poisson statistics with the use of Extreme Limiting Dilution Analysis (ELDA) online software (http://bioinf.wehi.edu.au/software/elda/). The $P$ indicates a statistically significant difference in TIC frequency between SAH-con or SAH-JGZ4. **l** Images of the patient-derived lung cancer organoids in 3D culture following vehicle or SAH-JGZ4 (5 μM) treatment for 7 days. Data are means ± SEM of three independent assays. Statistical significance between two groups was determined with two-tailed Student's $t$ test. Statistical significance among groups was determined by one-way ANOVA test. Source data are provided as a Source Data file.

EGFR degradation but also blocking the compensation of oncogenic or bypass signals.

As TRIB3 was reported to connect with NSCLC progression or response to therapeutic agents with some controversial opinions[41–45], we evaluated the genetic depletion of TRIB3 on

NSCLC progression. Knocking down of *TRIB3* inhibited tumor growth, metastasis, and increased the survival rate of tumor-bearing mice in NSCLC PDX and A549 xenograft models (Fig. 8l, m and Supplementary Fig. 8a–c), confirming the tumor-promotion role of TRIB3 in NSCLC. To identify which NSCLCs

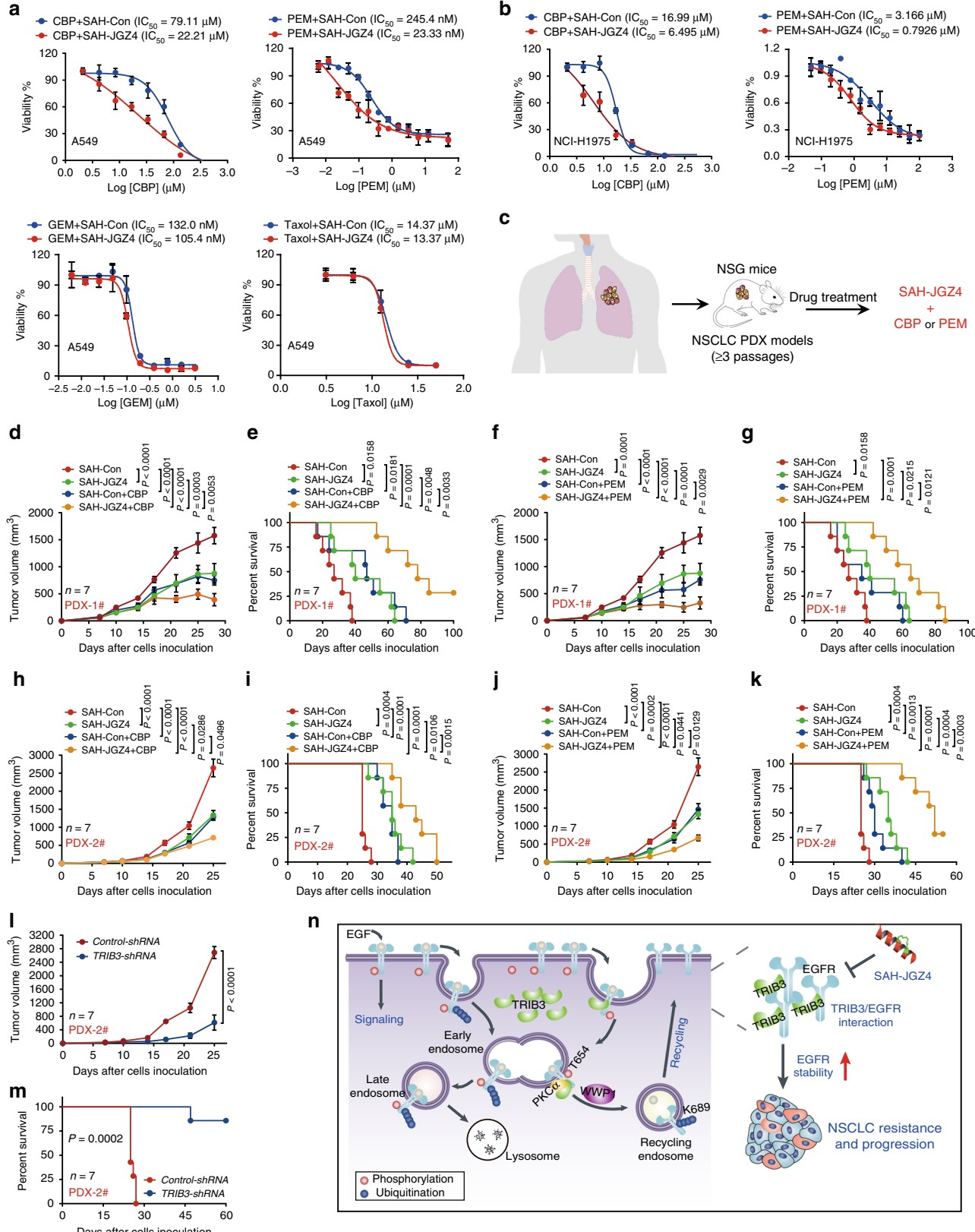

would be sensitive to SAH-JGZ4 treatment, three more NSCLC cell lines with different expression level of TRIB3 and EGFR (shown in Fig.1a) were tested using in vivo models. SAH-JGZ4 administration inhibited tumor growth of NCI-H460 cells but not NCI-H157 and NCI-H2170 cells (Supplementary Fig. 8d–f),

suggesting that tumors with high expression of both TRIB3 and EGFR would be sensitive to SAH-JGZ4 treatment. Consistently, SAH-JGZ4 treatment showed no further strengthened antitumor effects on *TRIB3* depletion cells (Supplementary Fig. 8a–c). We also evaluated the off-target and toxic effects of SAH-JGZ4. SAH-

**Fig. 8 SAH-JGZ4 sensitizes NSCLC to chemotherapy. a** Cell viability of A549 cells treated with different concentrations of carboplatin, pemetrexed, gemcitabine, or taxol in combination with 5 μM SAH-JGZ4 or SAH-Con for 72 h. Data are means ± SEM of three independent assays. **b** Cell viability of H1975 cells treated with different concentrations of carboplatin or pemetrexed in combination with 5 μM SAH-JGZ4 or SAH-Con for 72 h. Data are means ± SEM of three independent assays. **c** Schematic diagram for the construction of patient-derived tumor xenograft (PDX) models from NSCLC patients. Effects of carboplatin in combination with SAH-JGZ4 or SAH-Con on tumor growth (**d**) and survival (**e**) in the PDX-1# model. Effects of pemetrexed in combination with SAH-JGZ4 or SAH-Con on tumor growth (**f**) and survival (**g**) in the PDX-1# model. Effects of carboplatin in combination with SAH-JGZ4 or SAH-Con on tumor growth (**h**) and survival (**i**) in the PDX-2# model. Effects of pemetrexed in combination with SAH-JGZ4 or SAH-Con on tumor growth (**j**) and survival (**k**) in the PDX-2# model. Effects of TRIB3 deletion on tumor growth (**l**) and survival (**m**) in PDX-2# model. **n** Schematic diagram illustrates the molecular mechanism of how TRIB3 promoting EGFR recycling and stability, as well as NSCLC development. Statistical significance between two groups was determined with two-tailed Student's *t* test. Data in **d**, **f**, **h**, **j**, and **l** are presented as means ± SEM. Statistical significance among groups was determined by one-way ANOVA test. Statistical differences for the survival in **e**, **g**, **i**, **k**, and **m** were determined by two-sided log-rank test. CBP carboplatin, PEM pemetrexed, GEM gemcitabine. Source data are provided as a Source Data file.

JGZ4 did not affect the interaction of EGFR or TRIB3 with their known binding partners (Supplementary Fig. 9a, b and Supplementary Note 1). Another, SAH-JGZ4 showed a smaller impact on normal epithelial cells than on cancerous epithelial cells and revealed no abnormalities on all major organs (Supplementary Fig. 9c–i and Supplementary Note 1). Taken together, high expression of both TRIB3 and EGFR is the biomarker to determine the sensitivity of cancer cells to SAH-JGZ4 treatment.

## Discussion

Although targeted EGFR therapeutics such as TKIs are largely effective in the treatment of NSCLC with mutated EGFR, the development of resistance is a challenging issue for using TKI therapies in these patients[3]. Also, WT-EGFR is critically contributed to EGFR TKI resistance and NSCLC progression. WT-EGFR was reported to confer acquired resistance to third generation of EGFR TKIs and maintain the mutated KRAS activity as well as KRAS-driven tumorigenesis (another critical driving factor of NSCLC)[9,11,46]. Moreover, EGFR promotes tumor progression and therapeutic resistance independent of its kinase activity[47,48]. These evidences emphasize that WT-EGFR should be taken into account in both basic and translational researches. Full activation of EGFR, as well as termination of its signaling, depends on ligand-stimulated endocytosis and intracellular trafficking. Intrinsic and extrinsic stresses, including iatrogenic stress, trigger robust EGFR trafficking and signaling to provide cancer cells with a survival benefit and resistance to therapeutics[49]. However, the molecular mechanisms responsible for aberrant EGFR trafficking are far from elucidated. Data from our group and others suggest that TRIB3 is a stress sensor in response to a diverse range of stressors, allowing TRIB3 to participate in the pathogenesis of chronic inflammatory and malignant diseases by interacting with intracellular signaling and functional proteins[20,41,50–52]. In this study, we showed that the elevated TRIB3 participates in the pathogenesis and progression of NSCLC by enhancing EGFR recycling and stability, not only for the WT-EGFR, but also for the activating and "gatekeeper" mutants.

Mechanistically, TRIB3 interacts with EGFR and PKCα to form a heterotrimeric complex, in which TRIB3 acts as a scaffolding in recruiting PKCα to maintain a sustainable interaction with EGFR and elicit T654 phosphorylation of EGFR, a modification that blocks EGFR degradation[24,53]. Indeed, PKCα-induced T654 phosphorylation of EGFR acts as a marker for WWP1 to induce the K63-linked K689 ubiquitination of EGFR, a decisive signal for EGFR recycling rather than degradation. Although the majority of studies pay attention to the role of ubiquitination in mediating protein degradation, growing evidence suggests that ubiquitination, especially the K63-linked polyubiquitin modification, does not always serve as a degradation signal, but has roles in signal transduction, transcription, and other regulatory pathways[54]. K63-mediated B-cell lymphoma 6 (BCL6) polyubiquitination promotes BCL6 stabilization and lymphomagenesis[55]. For EGFR itself, SMURF2-induced ubiquitination enables EGFR stabilization[27]. Our data indicated that TRIB3–PKCα–WWP1 forms a positive regulatory axis on EGFR recycling and stability via inducing K63-linked ubiquitination of EGFR at K689. WWP1 belongs to the Nedd4-like homologous to the E6-associated protein C-terminus type E3 family, and its expression is upregulated across cancers[56]. In a previous study, WWP1 was identified to enhance EGFR expression by inhibiting the ring finger protein 11 activity[57]. The current work expands our understanding of ubiquitination in determining of protein fates, not only in degradation, but also for recycling and stability maintenance. In addition, our work shows molecular details about the tumor-promotion effect of WWP1 by inducing K63-linked K689 ubiquitination of EGFR, which subsequently promotes EGFR recycling and maintains EGFR stability. Crosstalk between different types of posttranslational modifications for precise and specific regulation is an emerging theme in eukaryotic biology[58]. Our study reveals that TRIB3 supports PKCα-induced T654 phosphorylation, which serves as a priming signal for the K689 ubiquitination of EGFR and elicits EGFR recycling and stabilization.

Except for EGFR per se, activation of alternative RTKs or STAT3 signaling is also critical for EGFR TKI or chemotherapy resistance in NSCLC[59,60]. High expression of TRIB3 is reported to correlate with poor response to erlotinib in NSCLC tumors that do not harbor the active EGFR mutations[43]. Such phenomenon may be the comprehensive effects of TRIB3 in promoting the stability not only of EGFR, but also of c-MET, the major bypass signaling pathway for EGFR TKI resistance[61]. In addition, TRIB3 is necessary for EGF-induced STAT3/5 activation, which is critical for cancer stemness and chemoresistance[59,62]. Here, we developed SAH-JGZ4, a staple-modified EGFR-derived peptide, which disturbs the TRIB3–EGFR interaction and produces anticancer effects by inhibiting EGFR recycling and subsequently inducing EGFR degradation. Moreover, SAH-JGZ4 treatment also showed inhibitory effects on the compensatory pathways by promoting c-Met degradation and sustaining inhibition of STAT3/5 signaling. The mutations of EGFR and KRAS are two main driver-alterations in NSCLC. In contrast to EGFR-targeted therapies, no effective inhibitor targeting mutated KRAS protein is available in the clinics. Recent study reveals that genetic ablation of EGFR suppresses KRAS activity and mutated KRAS-driven pathogenesis and progression in lung cancer[9,46]. Consistent with these observations, we found that SAH-JGZ4 treatment of KRAS mutated A549 cells decreases KRAS activity, and inhibits tumor initiation and progression, suggesting that promoting EGFR degradation have therapeutic potential for targeting the "undrugable" KRAS mutations. The therapeutic effect of SAH-JGZ4 in combination with KRAS inhibitors should be evaluated in future studies. For

TKI-based therapy, acquired resistance inevitably arises from the increasing affinity of the mutant receptor for ATP, which, in turn, diminishes the potency of these ATP-competitive inhibitors[12]. SAH-JGZ4 inhibits EGFR signaling via a mechanism entirely different from that of EGFR TKIs. Moreover, the active site targeted by SAH-JGZ4 is outside the tyrosine kinase domain but in the JM region of EGFR. These features of SAH-JGZ4 indicate that it is an effective and alternative agent for EGFR-targeted therapy, especially for overcoming the TKI resistance.

In current study, we performed one-round optimization of the original peptide based on its predictive α-helical conformation by using peptide stapling technique. The stapled SAH-JGZ4 displayed much better druggability, such as improved α-helicity, binding affinity, cytosolic penetration, and intracellular stability than the original peptide. The amino acids of Leu680, Ile682, Leu683, and Arg684 have been identified as the important positions for the peptide binding with TRIB3. Leu683 may be important for maintaining the binding conformation, as stapling on this site obtained a successful stapled peptide construct. Notably, although Ile682 appeared to be important as Leu680 and Arg684 for the peptide binding with TRIB3, Ile682 would be on the opposite face to Leu680 and Arg684 of an α-helix, suggesting that the peptide bound with TRIB3 not as a helix at some extent. Protein conformation is a dynamic and flexible status depending on the environmental factors (e.g., pH, ionic strength and temperature) and its binding molecules. From this aspect, using computational approach to predict the secondary structure of peptide has obvious limitation without taking such critical factors into account. Later, great effort should be made to determine the crystal structures of TRIB3 per se and TRIB3 in complex with EGFR or with the stapled peptide, which will provide valuable information not only for understanding the physiological and pathological roles of TRIB3; but also provide atomic evidence for therapeutic peptide design and optimization.

In summary, our studies suggest that the synergistic expression and action of TRIB3, EGFR, and PKCα establish a TRIB3–PKCα–WWP1 regulatory axis to promote NSCLC development by enhancing EGFR recycling, stability, and signaling. In addition, our work shows that targeting the TRIB3–EGFR interaction to promote EGFR degradation is a potential therapeutic option for the treatment of EGFR-related NSCLC cases (Fig. 8n).

## Methods

For detailed description of all methods, please see "Supplementary Methods".

**Cell lines and primary cultures**. Human NSCLC cell lines NCI-H1703, NCI-H2170, NCI-H157, NCI-H1395, NCI-H1975, A549, NCI-H1650, NCI-H460, NCM460, BEAS-2B, 4T1, and HaCaT were cultured in RPMI-1640 medium (GIBCO, Carlsbad, CA) supplemented with 10% fetal bovine serum (FBS). HEK 293T cells were cultured in IMDM medium (GIBCO, Carlsbad, CA) supplemented with 10% FBS. Primary lung cancer cells (Cell Biologics Inc, Chicago, USA) were cultured in complete human epithelial cell medium. All cells are maintained at 37 °C in a humidified 5% $CO_2$ atmosphere. NCM460 cells were obtained from GuangZhou Jennio Biotech Co., Ltd. 4T1 Cells were provided by Dr. Bo Huang from Institute of Basic Medicine, Chinese Academy of Medical Sciences & Peking Union Medical College. Other cell lines were obtained from the Cell Culture Center of Peking Union Medical College. No further authentication of these cell lines was performed. All of the cell lines were determined to be negative for mycoplasma using the MycoAlert Mycoplasma Detection Kit (Lonza, LT07-418). Cells were used for experiments within 15–20 passages from thawing.

**Mice**. For the xenograft experiments, five-week-old male BALB/c nude mice or NCG (NOD-Prkdc^scidIl2rg^null) mice were purchased from HFK Bioscience Co., Ltd (Beijing, China) and Nanjing Biomedical Research Institute of Nanjing University (Nanjing, China), respectively. Mice were maintained in the animal facility at the Institute of Materia Medica under specific-pathogen-free conditions. Mice were housed in groups of 4–6 mice per individually ventilated cage in a 12 h light/dark cycle (07:30–19:30 light; 19:30–7:30 dark), with controlled room temperature (23 ± 2 °C) and relative humidity (40–50%). For animal studies, the mice were earmarked before grouping, and then were randomly separated into groups by an independent person; however, no particular method of randomization was used. Sample size was predetermined empirically according to previous experience using the same strains and treatments. Generally, we used $n ≥ 6$ mice per group. We ensured that experimental groups were balanced in terms of animal age and weight. All animal studies were approved by the Animal Experimentation Ethics Committee of Chinese Academy of Medical Sciences, and all procedures were conducted in accordance with the guidelines of Institutional Animal Care and Use Committees of Chinese Academy of Medical Sciences. The animal study also accorded with the ARRIVE guidelines[63].

**Human subjects**. Lung cancer tissues were obtained from Cancer Institute and Hospital, Chinese Academy of Medical Science and the Guangdong provincial people's Hospital. Informed consent was obtained from all participants in accordance with the Declaration of Helsinki. All protocols using human specimens were approved by the Institutional Review Board of the Chinese Academy of Medical Sciences and Peking Union Medical College. The clinical features of the patients are listed in Supplementary Table 2 and Supplementary Table 3.

**Biotinylation and recycling assay of EGFR**. Cells were surface labeled with EZ-Link Sulfo-NHS-SS-Biotin (Thermo Scientific, Wilmington, DE, USA) on ice, and internalization was then allowed to proceed for 30 min at 37 °C with EGF (100 ng ml$^{-1}$). Biotin remaining at cell surface was removed by exposure to MesNa for three times (4 °C for 10 min). Overall, 1/4 cells were left as control and kept on ice to represent the total biotinylated EGFR. The remaining cells were divided into three equal parts and rewarmed to 37 °C in serum-free medium for indicated times, respectively, to allow recycling of the internalized EGFR. Cells were then re-exposed to MesNa for three times (4 °C for 10 min) to remove the Biotin conjugated with membrane EGFR. The total and the remaining biotinylated EGFR were determined by ELISA using microtiter wells coated with anti-EGFR Ab (Santa Cruz, sc-03, 1:500). "Recycling%" was calculated using the formula: Recycling% = (total − remaining)/total × 100%.

**Synthesis of stapled peptide**. Stapled SAH-JGZ peptides were formed by incorporating JGZ peptide (on the basis of the second α-helix in EGFR JM region) with two units of nonnatural alkenyl amino acids S5 at the relative positions $i$ and $i + 4$, and then cross-linked by ring-closing olefin metathesis, resulting in a stapled peptide that is braced in a α-helical conformation[64]. All stapled peptides, the chimeric peptide Pep2-JGZ, and their fluorescently labeled analogs were synthesized by Chinese peptide Company (Hangzhou, China). All peptides were purified by reverse-phase high performance liquid chromatography to >95% purity and quantitated by amino acid analysis.

**Quantification and statistical analysis**. Data are presented as the mean ± standard error. Statistical significance between two groups was determined with unpaired two-tailed Student's $t$ test. Statistical significance among groups was determined by one-way ANOVA test. The correlation between groups was determined by Pearson's correlation test. The survival rates were analyzed by the Kaplan–Meier method. Generally, all experiments were carried out with $n ≥ 3$ biological replicates. $P < 0.05$ was considered statistically significant. Analyses were performed using the Graphpad Prism 7.0 software.

**Reporting summary**. Further information on research design is available in the Nature Research Reporting Summary linked to this article.

## Data availability

All microarray data generated in this study have been deposited at the NCBI Gene Expression Omnibus with the accession code GSE103891. Correlation between EGFR and TRIB3 mRNA expression across TCGA lung cancer data sets were analyzed on the following website: http://gepia.cancer-pku.cn. The KM plotter lung cancer dataset was obtained from http://kmplot.com/analysis. All other data supporting the findings of this study are available from the corresponding authors upon reasonable request. A Reporting Summary for this Article is available as a Supplementary Information file. The uncropped gel or blot figures and original data underlying Figs. 1–8 and Supplementary Figs. 1–9 are provided as a Source Data file. Source data are provided with this paper.

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

## Acknowledgements

We are grateful to Prof. Mien-Chie Hung's laboratory for the gifts of pcDNA6A-EGFR WT, pcDNA6A-EGFR ECD (1-644), and pcDNA6A-EGFR ICD (645-1186) plasmids. This work was supported by grants from the National Key R&D Program of China (2017YFA0205400), the National Natural Science Foundation of China (81773781 and 81530093 to Z-.W.H.; 81472717, 81673474 and 81973344 to F.H.; 81400140 to K.L.; and 81703564 to J-.j.Y.), Beijing Natural Science Foundation (7162133 to F.H.), CAMS Innovation Found for Medical Sciences (2016-I2M-1-007 to Z-.W.H. and F.H.; 2019-I2M-1-005 to F.H.; 2016-I2M-3-008 to B.C., J-.j.Y., F.W., S-.s.L., and J-.m.Y.; 2016-I2M-1-011 to K.L.) and Beijing Outstanding Young Scientist Program (BJJWZYJH01201910023028).

## Author contributions

Z-.W.H. and F.H. conceived and participated in the overall design, supervision, and coordination of the study. F.H., J-.j.Y., and D-.d.Z. designed and performed most of experiments as well as data analysis. X-.x.Y. performed chemical analysis study. F-.w.T. and J.W. collected the clinical samples and established the PDX models. B.C., K.L., X-.x.L., S-.s.L., and B.H. participated in the molecular and cellular biological experiments. S.S., C.Z., J-.m.Y., X-.w.Z., and F.W. participated in the animal studies. Z-.W.H., F.H., and J-.j.Y. wrote the manuscript. All authors read and approved the manuscript.

## Competing interests

The authors declare no competing interests.
