## [Peer Review File · Nature Communications]

Reviewers' comments:

Reviewer #1 (Remarks to the Author): expert in Tribble:

This is an excellent body of work with novel findings that will significantly impact research understanding and therapeutic targeting of NSCLC. The work shows a novel regulatory axis of TRIB3, PKCa, WWP1 and EGFR in NSCLC tumour maintenance, progression and stemness. TRIB3 interacts with EGFR, recruiting PKCa leading to the phosphorylation of EGFR at T654, which primes EGFR for WWP1-catalysed K63 linked K689 ubiquitination of EGFR, signaling for the recycling via endosomes of EGFR back to the cell membrane, resulting in its enhanced stability. This is a well executed and well controlled study.

I have the following comments:

1. In the introduction, it is mentions the EGFR drugs rociletinib and AZD9291 and cetuximab. A more thorough synopsis of the EGFR drugs used in NSCLC should be providing in the introduction, and what they target (single, dual or pan-EGFR targets). Gefitinib (used in this study) is a first generation EGFR single inhibitor. AZD9291 otherwise know as osimertinib is a dual mutant EGFR and Her2 targeting drug as there is evidence for efficacy in Her2 positive patients (Liu et al, Clin Cancer res 2018), and rociletinib against mutant EGFR. Given that the data shows that the peptide has superior activity to gefitinib, but similar efficacy to AZD9291, but does not affect ERBB2 (Her2) it raises this question: Does AZD9291 act on the regulatory axis identified in this study? Does it affect TRIB3/PKCa/WWP1/EGFR recycling? This would be very interesting to know, or does AZD9291 affect ERBB2 whilst the peptide does not. I think this would be important to clarify how the peptide and AZD9291 have similar efficacy, as shown by the authors. Also, can you comment on whether you know gefitinib does not affect TRIB3, explaining its lack of efficacy?
2. Why not use consistent terminology with the literature that support the "scaffolding" role of TRIB proteins which have long been postulated, or are you making a distinction between a scaffold and a buckle, if so please expand in the text? To solidify the terminology for TRIB protein function as a scaffold, I would recommend the use of the "scaffolding" term rather than variety of terms i.e. "buckle" or Universal molecular connector" "molecular buckle" which the authors use.
3. Figure 3f-j not referred to in text showing the specific domains involved in interaction between TRIB3 and EGFR, PKC and TRIB3 and rescue of recycling with overexpression of M5 only of TRIB3. This is really nice data and worthy of attention in the body of text.
4. Does the half life increase in the T654A mutant? It shows that it recycles as fast as WT in the presence of TRIB3, so assuming the authors are correct, its stability in the absence of TRIB3 should be higher that WT.

Text/figure edits:

5. The use of abbreviations throughout the paper should be checked that they are all spelled out on first mention. Names of genes should be named in full also on first mention e.g. SC4MOL, NSDHL, GOLM1 in introduction, TIC in figure 7 legend/text.
6. For figure 1a, What does means +-SEM of 3 independent assays mean. By assays do you mean western blots and if so that should be specified and if calculated by densitometry or what method. The legend should be more precise. Is this detailed in the methods?
7. It would be recommended to use a different color for untreated in figure 2d so its clearer that there is reduction in cell surface expression in knockdown without inhibitor treatment.
8. How is figure 2d calculated, please specify in legend.
9. In figure 3a, it should be explained and said in the legend what the positive and negative controls are in this screen.
10. The figure in supplemental S4g nicely support (by eye ball of slides) the stated results but there is something mislabelled in the associated graph as the mutant labels do not match.

Reviewer #2 (Remarks to the Author); expert in stapled peptides:

This manuscript describes a comprehensive study that covers multiple disciplines ranging from complex biochemical analyses in vitro and in vivo, over peptide antagonist engineering, to in vivo pharmacological analysis of the antagonist. Based on my background in peptide engineering, I am reviewing here exclusively the part that concerns the engineering and in vitro evaluation of the stapled peptide TRIB3/EGFR protein-protein interaction inhibitor (PPI)

The strategy of inhibiting the protein-protein interaction of TRIB3/EGFR by a stapled peptide is well chosen as this approach has been working for a number of examples in which an alpha helix at the interface two proteins was mimicked by a synthetic peptide stabilized in it's a-helical conformation by a hydrocarbon staple. The approach can offer rapid access to valuable tool compounds. Care needs to be taken concerning the cell permeability of stapled peptides as the membrane permeability of peptides of this size and polarity can be low. Concerning the stapled peptide engineered in this study by Hu an co-workers, I have some concerns regarding the i) the design of

the stapled peptides, ii) their binding affinity, and iii) their cell permeability, as discussed in detail in the following.

1. Design of stapled peptides:

The authors have predicted the regions in EGFR that form α -helices and bind TRIB3 by a computational tool, which is risky as the prediction may be wrong. Other studies developing stapled peptides have based their designs on X-ray structures of protein-protein interactions, which is safer. It would be a big plus if the authors demonstrate experimentally that the chosen peptide region indeed forms a helix that binds to TRIB3, although this might need a substantial effort (e.g. by NMR, X-ray).

A point concerning the peptide design that was not clear to me is why the peptide JMA2 (linear wt sequence) is not binding TRIB3 but the mutant JGZ that has a Lys \rightarrow Arg mutation apparently does. The rationale for the Lys \rightarrow Arg mutation is not obvious to follow for a peptide engineer. Did the authors try many other mutations and was this one a "lucky" finding?

The authors have then chosen amino acid positions i and $i+4$ to introduce unnatural amino acids for stapling the putative helices, as usually done to generate stapled peptides. As they did not have structural data and could not predict which amino acid positions/face of the helix would tolerate a staple, they had made four variants (see Fig. 6e, lower panel). For two of the four constructs, stapling amino acids 3,7 and 6, 10, they report nanomolar binding affinities. In my view, it is nearly impossible that both the two bind, as this means that amino acids 3, 7, 6 and 10 are at a face that does not interact with TRIB3. This leaves not many amino acids at an opposite face of the α -helix that can bind. Structure determination by X-ray or NMR would help to clarify this situation (although this needs a large effort). An alanine scan of the peptide JGZ could identify the amino acids that are key for the interaction and would tell if there are amino acids at the opposite face that interact with TRIB3.

2. Binding affinity

The authors determined the binding of the peptides (linear, stapled) only with one technique, SPR, and the presented SPR data is somehow conflicting for the following reasons. The dissociation constants reported were calculated based on the association and dissociation rates of the SPR curves. However, this data does not fit with the steady-state data that can be extracted from the SPR curves shown in Figs. 6b and 6e. At increasing concentrations of peptide injected into the SPR chip, the response is still increasing substantially, even at micromolar concentrations. This is not expected for a nM binder. If the dissociation constants are calculated based on this steady state data, one would obtain micromolar K_d s and thus much weaker binders than reported. The authors should calculate the K_d s also based on the steady state data and analyze the reasons for the discrepancy

with their reported values. In addition, I strongly recommend to measure the binding affinity with a second, independent method. Given that the authors have the peptides also labeled with fluorescein, they could easily measure the K_d s by fluorescence polarization, which is a robust method. In addition, they could do fluorescence polarization competition measurements to show that the peptide can displace the TRIB3/EGFR interaction.

3. Cell permeability

The target of the stapled peptide, TRIB3 is inside the cell and the peptides thus need to pass the cell membrane. The authors claim that the peptide is cell permeable but they have used a technique that is, in my view, not suited to assess the cell permeability (and not accepted in the field). Specifically, they analyzed it by FACS, quantifying the fluorescence of cells that were incubated with the fluorescein-labeled peptide. The problem is that FACS shows also fluorescence in endosomes or other compartments and it is not clear if the peptides have entered the cytosol. I recommend that the authors quantify the concentration of stapled peptide reached in the cytosol with an accepted method. A highly suited method is for example the quantification by the Halo Tag (labeling of peptide with haloalkane and capture in cells with the Halo tag).

In summary, I have three major concerns that I think need to be addressed all experimentally, namely

i) Design of peptide: confirm interaction of stapled peptide with TRIB3 by structure (NMR may be easiest), or show specificity with an alanine scan

ii) Binding affinity: measure K_D with alternative method, e.g. fluorescence polarization, show also reversibility, and show that the peptide can disrupt the TRIB2/EGFR interaction in vitro

iii) Demonstrate delivery of peptide into cytosol with established method (e.g. halo tag approach)

Again, please note that my evaluation is limited to the chapter about the engineering and in vitro characterization of the stapled peptide, and I do not comment on any of the other parts of the work.

Reviewer #3 (Remarks to the Author); expert in NSCLC and EGFR-TKI resistance:

NCOMMS-19-15423-T

Hua et al. "TRIB3/EGFR Interaction promotes lung cancer progression and defines a therapeutic target "

The authors continue their study of TRIB3 and focus on its role in non small cell lung cancer (NSCLC) and EGFR. Their basic finding is that TRIB3 contributes to EGFR protein stability through direct interaction and EGFR and PKCalpha. Knockdown TRIB3 leads to decrease in EGFR protein and phosphorylation. Overall they present data claiming that TRIB3 blocks EGFR degradation and thus "targeting" TRIB3 should lead to an anti-EGFR effect. They develop a "stapled" peptide to block TRIB3 EGFR interaction and show that it inhibits growth of NSCLC cell lines in vitro and in xenograft models, inhibits cancer stem cell like properties (re-transplantation), and metastatic behavior. Thus, this peptide can be tested in vivo in preclinical models. This occurs in EGFR wild-type and EGFR mutant tumors. They also present data that high protein expression of TRIB3 is associated with poor prognosis, that stapled peptide targeting TRIB3 leads to better responses to cisplatin. In all of these things they emphasize this works in EGFR wild type as well as EGFR mutant NSCLCs. They conclude: "Disturbing TRIB3/EGFR interaction with a stapled peptide attenuates initiation and progression of TKIs sensitive and -resistant NSCLCs by accelerating EGFR degradation. These findings indicate that targeting EGFR degradation is a previously unappreciated therapeutic option against EGFR-related NSCLC. " No data on development of resistance to TRIB3 "stapled peptide" targeted therapy are provided.

Comments to the Editors:

This paper is reviewed in the context of the urgent need to have additional targeted therapies for NSCLC particularly for NSCLC expressing EGFR wild-type protein, and to aid in developing much deeper and potentially curative therapies for EGFR mutant tumors. In addition, the paper is reviewed in the context of several other TRIB3 publications by the authors and others both for lung cancer, and mechanistically showing a role in degradation of potential oncoproteins. The authors provide very large dataset for multiple aspects of their experiments. There are several problems with the current work.

1. The quantitative effect of their targeted therapy. While they have statistically significant effects in vitro and in vivo these do not appear to be quantitatively dramatic or provide long term disease free survival. While the data are what the data are, at the end of the day for clinical translation we need to ask, based on the preclinical data would this be a "game changer"? From what I see it does not appear to be that.

2. As part of this game changing approach combination therapies could be tried and I would be happy if the TRIB3 targeting provided the “final push” to cure/long term survival – but such data are lacking.

3. Does resistance to the stapled peptide occur? Clearly in all other EGFR targeted therapies dealing with resistance is a central issue, and we equally have to know this about targeting TRIB3.

4. While the stapled peptide is one approach, what about a comparison to simply genetically removing TRIB3 and determining the effects on xenograft growth and metastases as well as the potential to develop resistance even with TRIB3 missing altogether? One reason I say that, comes from data in the “Dependency Map” (DepMap) which shows in siRNA and CRISPR studies that TRIB3 does not appear to be an essential gene by dropout looking at a large panel of human cancer lines including NSCLC lines. The authors should look at and discuss the DepMap data as well as performing their own studies.

5. The authors should also look at KM Plotter – where they will see that high expression of TRIB3 mRNA is very significantly associated with inferior survival in NSCLC, and that this appears to be mainly in adenocarcinoma and not squamous lung cancer. The authors need to resolve these issues, since their current claims cut across all histologies of NSCLC.

6. The effect of the stapled peptide on cisplatin response shows only a modest 3 fold or so sensitization. It would be useful (whatever the results are) to know the results for other chemotherapies routinely used in NSCLC treatment such as pemetrexed, taxanes, gemcitabine alone or in combination with platin (as given in the clinic. While there may or may not be sensitization a paper with these claims needs to define these data. Obviously if it acts as significant sensitizer – great. If not, at least we know.

7. I did a simple literature search and found there is significant literature on TRIB3 in lung cancer that I believe was not cited by the authors (see below). While they do not have to study all of these, given their very specific mechanistic claims, clearly it would be important to know what role if any these other mechanisms play, how quantitatively significant they are, and would one want to use both stapled peptide and these other targets. Some obvious ones that are easy to test experimentally include emodin, salinomycin, and ABTL0812 (which actually claims to work by up regulating TRIB3!). Related to this is what happens to notch1 with TRIB3 targeting given the Mol Med Rep 2013 used similar NSCLC preclinical models? Now it may be the authors know about these studies in detail, and don't find them compelling. However, given their claims in this paper it would seem prudent for them to at least discuss these other papers.

Ding CZ, Guo XF, Wang GL, Wang HT, Xu GH, Liu YY, Wu ZJ, Chen YH, Wang J, Wang WG. High glucose contributes to the proliferation and migration of non-small cell lung cancer cells via GAS5-TRIB3 axis. Biosci Rep. 2018. PMID: PMC5857909.

Su J, Yan Y, Qu J, Xue X, Liu Z, Cai H. Emodin induces apoptosis of lung cancer cells through ER stress and the TRIB3/NF-kappaB pathway. Oncol Rep. 2017;37(3):1565-1572. PubMed: 28184934.

Snezhkina AV, Krasnov GS, Zaretsky AR, Zhavoronkov A, Nyushko KM, Moskalev AA, Karpova IY, Afremova AI, Lipatova AV, Kochetkov DV, Fedorova MS, Volchenko NN, Sadritdinova AF, Melnikova NV, Sidorov DV, Popov AY, Kalinin DV, Kaprin AD, Alekseev BY, Dmitriev AA, Kudryavtseva AV. Differential expression of alternatively spliced transcripts related to energy metabolism in colorectal cancer. *BMC Genomics*. 2016;17(Suppl 14):1011. PMID: PMC5249009.

Erazo T, Lorente M, Lopez-Plana A, Munoz-Guardiola P, Fernandez-Nogueira P, Garcia-Martinez JA, Bragado P, Fuster G, Salazar M, Espadaler J, Hernandez-Losa J, Bayascas JR, Cortal M, Vidal L, Gascon P, Gomez-Ferreria M, Alfon J, Velasco G, Domenech C, Lizcano JM. The New Antitumor Drug ABTL0812 Inhibits the Akt/mTORC1 Axis by Upregulating Tribbles-3 Pseudokinase. *Clin Cancer Res*. 2016;22(10):2508-19. PubMed: 26671995.

Lopez-Ayllon BD, de Castro-Carpeno J, Rodriguez C, Pernia O, Ibanez de Caceres I, Belda-Iniesta C, Perona R, Sastre L. Biomarkers of erlotinib response in non-small cell lung cancer tumors that do not harbor the more common epidermal growth factor receptor mutations. *Int J Clin Exp Pathol*. 2015;8(3):2888-98.: PMC4440106.

Li T, Su L, Zhong N, Hao X, Zhong D, Singhal S, Liu X. Salinomycin induces cell death with autophagy through activation of endoplasmic reticulum stress in human cancer cells. *Autophagy*. 2013;9(7):1057-68. PMID: PMC3722315.

Zhou H, Luo Y, Chen JH, Hu J, Luo YZ, Wang W, Zeng Y, Xiao L. Knockdown of TRB3 induces apoptosis in human lung adenocarcinoma cells through regulation of Notch 1 expression. *Mol Med Rep*. 2013;8(1):47-52. PubMed: 23632994.

8. The authors need to state whether TRIB3 is mutated or amplified in NSCLC. Data easily available from TCGA

9. How do the authors propose to identify which NSCLCs would be sensitive to TRIB3 targeted therapy? Would this be by EGFR and TRIB3 levels or something else? Clearly data on a larger panel of NSCLC preclinical models would show whether there are some obvious biomarker correlations.

10. Substantial editing of English grammar is needed.

Reviewer #1 (Remarks to the Author): expert in Tribble:

This is an excellent body of work with novel findings that will significantly impact research understanding and therapeutic targeting of NSCLC. The work shows a novel regulatory axis of TRIB3, PKC α , WWP1 and EGFR in NSCLC tumour maintenance, progression and stemness. TRIB3 interacts with EGFR, recruiting PKC α leading to the phosphorylation of EGFR at T654, which primes EGFR for WWP1-catalysed K63 linked K689 ubiquitination of EGFR, signaling for the recycling via endosomes of EGFR back to the cell membrane, resulting in its enhanced stability. This is a well executed and well controlled study.

I have the following comments:

1. In the introduction, it is mentions the EGFR drugs rociletinib and AZD9291 and cetuximab. A more thorough synopsis of the EGFR drugs used in NSCLC should be providing in the introduction, and what they target (single, dual or pan-EGFR targets). Gefitinib (used in this study) is a first generation EGFR single inhibitor. AZD9291 otherwise know as osimertinib is a dual mutant EGFR and Her2 targeting drug as there is evidence for efficacy in Her2 positive patients (Liu et al, Clin Cancer res 2018), and rociletinib against mutant EGFR. Given that the data shows that the peptide has superior activity to gefitinib, but similar efficacy to AZD9291, but does not affect ERBB2 (Her2) it raises this question: Does AZD9291 act on the regulatory axis identified in this study? Does it affect TRIB3/PKC α /WWP1/EGFR recycling? This would be very interesting to know, or does AZD9291 affect ERBB2 whilst the peptide does not. I think this would be important to clarify how the peptide and AZD9291 have similar efficacy, as shown by the authors. Also, can you comment on whether you know gefitinib does not affect TRIB3, explaining its lack of efficacy?

RE: We thank this reviewer for his encourage comments regarding our study. Following your valuable and constructive suggestions, we conduct new experiments, modify or correct our MS. Particularly, we have re-written the section of Introduction, where we briefly described several common EGFR inhibitors, discussed how TKIs and mAbs inhibit EGFR signaling, and introduced the targets for each generation of EGFR TKIs. However, because EGFR-targeted antibodies are mainly used for advanced colorectal and head & neck cancers but not NSCLCs, we did not give much space to introduce them.

To determine if AZD9291 regulated the TRIB3/PKC α /WWP1/EGFR axis, we detected the expression of pEGFR^{T654} and total EGFR under AZD9291 treatment. In H1975 cells, AZD9291 treatment for 12 hr markedly reduced pEGFR^{Y1068} but had no effect on the expression of pEGFR^{T654}, PKC α , TRIB3 and EGFR as SAH-JGZ4 did. AZD9291 inhibited HER2 phosphorylation but SAH-JGZ4 did not (Data presented as Fig. S7g in revised MS). We examined the *in vivo* long-term effect of SAH-JGZ4 and AZD9291 on EGFR downstream signaling in H1975 xenograft tumor samples. Both SAH-JGZ4 and AZD9291 treatment decreased pEGFR^{Y1068}. Interestingly, differing from the short-term effect, one-month AZD9291 treatment markedly reduced EGFR and TRIB3 expression,

as well as slight reduction of pEGFR^{T654} (presented as Fig. S7d in revised MS). However, AZD9291 had no effect on EGFR-TRIB3 interaction as SAH-JGZ4 did (Fig. R1A). As EGF stimulation induces TRIB3 expression (Fig. R1B), long-term AZD9291 treatment may indirectly suppress the TRIB3/PKC α /EGFR axis via reducing TRIB3 expression. Notably, long-term AZD9291 treatment markedly enhanced the basal and phosphorylation of STAT3, STAT5 and ERK1/2 (presented in Fig. S7d of the revised manuscript), suggesting a compensatory activation of other oncogenic pathways. Such compensatory activation was not observed in SAH-JGZ4 group. Thus, the peptide and AZD9291 show similar efficacy on H1975 tumor growth via differential molecular mechanisms. On one hand, AZD9291 is a dual inhibitor of EGFR/HER2 and may suppress PKC α /WWP1/EGFR axis in an indirect way. On the other hand, long-term AZD9291 treatment activates STAT3/5 and ERK signaling, which may support the lung cancer stemness. SAH-JGZ4 promotes EGFR degradation to attenuate EGFR signaling and maintain a long-term tumor stemness inhibition, but had no effect on HER2 blockade.

Notably, short- or long- term Gefitinib treatment had no effect on EGFR phosphorylation and expression in H1975 (presented as Fig. S7d in revised MS and showed below as R1C). However, long-term Gefitinib treatment markedly induced phosphorylation of STAT3 and STAT5 (presented as Fig. S7d in revised MS), indicating the compensatory activation of other oncogenic pathways. Thus, lacking efficacy of Gefitinib on H1975 cells is due to T790M-related resistance and compensatory activation of other oncogenic pathways. We have described and discussed these in the revised MS.

Figure R1.

2. Why not use consistent terminology with the literature that support the "scaffolding" role of TRIB proteins which have long been postulated, or are you making a distinction between a scaffold and a buckle, if so please expand in the text? To solidify the terminology for TRIB protein function as a scaffold, I would recommend the use of the "scaffolding" term rather than variety of terms i.e. "buckle" or Universal molecular connector" "molecular buckle" which the authors use.

RE: Following your suggestions, we have replaced the word "buckle" as "scaffolding" and made them consistent in the whole manuscript.

3. Figure 3f-j not referred to in text showing the specific domains involved in interaction between TRIB3 and EGFR, PKC and TRIB3 and rescue of recycling with overexpression of M5 only of TRIB3. This is really nice data and worthy of attention in the body of text.

RE: Sorry for our carelessness! The description referred to Figure 3f-j has been added back to the revised manuscript.

4. Does the half life increase in the T654A mutant? It shows that it recycles as fast as WT in the presence of TRIB3, so assuming the authors are correct, its stability in the absence of TRIB3 should be higher than WT.

RE: There may be some misunderstanding. In fact, our data showed that T654A together with or without TRIB3 recycled as fast as WT-only, but much slower than that of WT together with TRIB3. That means when Thr654 was mutated as alanine, EGFR recycling lost the chance to be accelerated by TRIB3. From this point, the stability of T654A mutant should not be higher than that of WT in the absence of TRIB3. We conducted CHX assay to detect the stability of EGFR-T654A and EGFR-WT (Fig. R2). Overexpression of TRIB3 enhanced the stability of stability of EGFR-T654A and EGFR-WT. Overexpression of TRIB3 prolonged the half-life of EGFR-WT from 6.895 hr to more than 24 hr. However, the T654A mutant showed shorter half-life than the WT, either in the absence or presence of TRIB3 (3.188 hr and 3.919 hr, respectively).

Figure R2.

Text/figure edits:

5. The use of abbreviations throughout the paper should be checked that they are all spelled out on first mention. Names of genes should be named in full also on first mention e.g. SC4MOL, NSDHL, GOLM1 in introduction, TIC in figure 7 legend/text.

RE: Following your suggestion, we have checked all the abbreviations (including the gene names) throughout the paper and spelled them out on first mention.

6. For figure 1a, what does means +/-SEM of 3 independent assays mean. By assays do you mean western blots and if so that should be specified and if calculated by densitometry or what method. The legend should be more precise. Is this detailed in the methods?

RE: We thank this reviewer's suggestions. Here the "3 independent assays" means three independent biological studies; and the western blots were quantified by densitometry and calculated relative to GAPDH. We have modified the legend of Figure 1a to provide a detailed description to show how the western blots was normalized. The statistical

analysis of immune-blotting has been described in the Methods (“Immunoblotting, Immunostaining and Immunohistochemistry” part).

7. It would be recommended to use a different color for untreated in figure 2d so its clearer that there is reduction in cell surface expression in knockdown without inhibitor treatment.

RE: Following your suggestion, we have changed the colors to make different groups much clearer to be distinguished. The modified graph is presented in the revised manuscript as Fig. 2d.

8. How is figure 2d calculated, please specify in legend.

RE: Following your criticism, we have modified the legend of Figure 2d to provide information showing how the cell surface EGFR was quantified and normalized.

9. In figure 3a, it should be explained and said in the legend what the positive and negative controls are in this screen.

RE: Following your suggestion, we have modified figure 3a (presented as Fig. 3a in the revised MS) and provided experimental details in the legend. Briefly, IgG labelled with Alexa-647 and Biotin-labelled BSA was used as positive controls in the screening. GST protein, the buffer only and BSA were used as negative controls.

10. The figure in supplemental S4g nicely support (by eye ball of slides) the stated results but there is something mislabelled in the associated graph as the mutant labels do not match.

RE: This was a mistake. We have corrected the labels. The modified graph is presented as Fig. S4g in the revised MS.

Reviewer #2 (Remarks to the Author); expert in stapled peptides:

This manuscript describes a comprehensive study that covers multiple disciplines ranging from complex biochemical analyses in vitro and in vivo, over peptide antagonist engineering, to in vivo pharmacological analysis of the antagonist. Based on my background in peptide engineering, I am reviewing here exclusively the part that concerns the engineering and in vitro evaluation of the stapled peptide TRIB3/EGFR protein-protein interaction inhibitor (PPI)

The strategy of inhibiting the protein-protein interaction of TRIB3/EGFR by a stapled peptide is well chosen as this approach has been working for a number of examples in which an alpha helix at the interface two proteins was mimicked by a synthetic peptide stabilized in its α -helical conformation by a hydrocarbon staple. The approach can offer rapid access to valuable tool compounds. Care needs to be taken concerning the cell permeability of stapled peptides as the membrane permeability of peptides of this size and polarity can be low. Concerning the stapled peptide engineered in this study by Hu and co-workers, I have some concerns regarding the i) the design of the stapled peptides, ii) their binding affinity, and iii) their cell permeability, as discussed in detail in the following.

1. Design of stapled peptides:

1) The authors have predicted the regions in EGFR that form α -helices and bind TRIB3 by a computational tool, which is risky as the prediction may be wrong. Other studies developing stapled peptides have based their designs on X-ray structures of protein-protein interactions, which is safer. It would be a big plus if the authors demonstrate experimentally that the chosen peptide region indeed forms a helix that binds to TRIB3, although this might need a substantial effort (e.g. by NMR, X-ray).

RE: We thank the reviewer's valuable comments. Firstly, only the secondary structure of the EGFR JM region was predicted with the computational tool, I-TASSER. This prediction was based on the PDB data (1Z9I) from an NMR analysis of EGFR JM region, which indicated that the JMA2 region in our manuscript is a α -helix (*J Biol Chem.* 2005, 280:24043-52). Secondly, the binding region of EGFR with TRIB3 was determined through CO-IP assays (Fig. 3h, 3i, 6a right) instead of prediction. Thirdly, obtaining the co-crystal structure of stapled peptide bound to its target can accurately show the binding interface of the peptide with its target. However, no crystal structure is available for TRIB3 as yet. We have attempted to do some work about it but failed because of some technical issues, especially the solubility of the recombinant TRIB3 protein. In the future, great effort should be made to determine the crystal structures of TRIB3 *per se* and TRIB3 in complex with its binding partners, which will provide valuable information for 1) understanding the physiological and pathological roles of TRIB3; 2) structure-based drug design and optimization. We have briefly discussed the importance of deciphering the crystal structure of TRIB3 alone or in complex with EGFR or with the stapled peptide in the revised MS.

2) A point concerning the peptide design that was not clear to me is why the peptide JMA2 (linear wt sequence) is not binding TRIB3 but the mutant JGZ that has a LysArg mutation apparently does. The rationale for the LysArg mutation is not obvious to follow for a peptide engineer. Did the authors try many other mutations and was this one a "lucky" finding?

RE: The I-TASSER analysis showed that L683 and K684 in the JMA2 amino acid sequence made the α -helix discontinuous. In fact, we tried three amino acid substitutions. In this study, amino acid with similar physicochemical properties was chosen for the replacement. Leucine is one of the branched-chain amino acids (leucine, isoleucine and valine). The leucine was substituted with isoleucine or valine, respectively. However, the two substitutions did not show acceptable binding affinity. Then the LysArg substitution was performed, because they are both basic amino acids with similar structure than that of histidine, another basic amino acid including an aromatic ring. Fortunately, this arginine substitution peptide (JMA2-S3, renamed as JGZ later) displayed a good binding affinity with TRIB3 ($K_D = 3.494E-07$ M) and was chosen for further stapled modification. The modified graph is presented as Fig. 6b in revised MS.

3) The authors have then chosen amino acid positions i and $i+4$ to introduce unnatural amino acids for stapling the putative helices, as usually done to generate stapled peptides. As they did not have structural data and could not predict which amino acid positions/face of the helix would tolerate a staple, they had made four variants (see Fig. 6e, lower panel). For two of the four constructs, stapling amino acids 3,7 and 6, 10, they report nanomolar binding affinities. In my view, it is nearly impossible that both the two bind, as this means that amino acids 3, 7, 6 and 10 are at a face that does not interact with TRIB3. This leaves not many amino acids at an opposite face of the α -helix that can bind. Structure determination by X-ray or NMR would help to clarify this situation (although this needs a large effort). An alanine scan of the peptide JGZ could identify the amino acids that are key for the interaction and would tell if there are amino acids at the opposite face that interact with TRIB3.

RE: Here, the reviewer may make a clerical error. In fact, 3,7 stapling lost the binding affinity; while 4,8 and 6,10 got the acceptable binding affinities. As there is some difficulty in determination the co-crystal structure of TRIB3 in complex with SAH-JGZ4 peptide by X-ray or NMR at this stage, we performed the alanine scanning to identify the key amino acids for the interaction. When amino acids on 5,7,8,9, or 14 sites was mutated as alanine, the resulted peptide totally lost the binding activity with TRIB3. Analyzing in combination with the stapled peptide data (Fig. 6e), the 6,10 amino acid positions of the helix should be tolerated the stapled modification. As each helical turn consists of about 3.6 amino acid residues, the 3,7 positions or the 5,9 positions may be the binding interface with TRIB3. It is worth noting that from the stapled peptide data, the 4,8 positions may also tolerate the stapled modification. However, alanine substitution of the 8th amino acid resulted in the loss of binding activity of the peptide. The 8th amino acid—L683 locates at the discontinuous α -helical site, the alanine substitution may cause the destruction of the α -helical structure and result in decrease in the binding affinity. The alanine scanning data has been illustrated as Fig. S5a in the revised MS.

2. Binding affinity

The authors determined the binding of the peptides (linear, stapled) only with one technique, SPR, and the presented SPR data is somehow conflicting for the following reasons. The dissociation constants reported were calculated based on the association and dissociation rates of the SPR curves. However, this data does not fit with the steady-state data that can be extracted from the SPR curves shown in Figs. 6b and 6e. At increasing concentrations of peptide injected into the SPR chip, the response is still increasing substantially, even at micromolar concentrations. This is not expected for a nM binder. If the dissociation constants are calculated based on this steady state data, one would obtain micromolar Kds and thus much weaker binders than reported. The authors should calculate the Kds also based on the steady state data and analyze the reasons for the discrepancy with their reported values. In addition, I strongly recommend to measure the binding affinity with a second, independent method. Given that the authors have the peptides also labeled with fluorescein, they could easily measure the Kds by fluorescence polarization, which is a robust method. In addition, they could do fluorescence polarization competition measurements to show that the peptide can displace the TRIB3/EGFR interaction.

RE: Following your constructive suggestions, we repeated the SPR assays and the Kd was re-calculated based on the association and dissociation rates of the SPR curves, as well as the steady state data. As shown in Fig. 6e and Fig. S5b in the revised MS, the Kds of SAH-JGZ4 calculated from the association and dissociation rates, or the steady state data were 6.615E-08 M and 1.369E-07 M, respectively. Following the strategy, all the SPR data were re-calculated and the values were modified in the revised version (presented as Fig. 6e in revised MS). Moreover, the fluorescence polarization (FP) assay was performed and the Kd value was calculated as 5.541E-08 M in this assay, the same magnitude as that from the SPR assay (presented as Fig. S5c in revised MS). Also, the unlabeled peptide corresponding to FAM-SAH-JGZ4 dose-dependently displaced the tracer peptide in the FP competitive assay, with a Ki value of 2.766E-08 M, a value similar to the apparent Kd derived from the FP binding assay (presented as Fig. S5d in revised MS). These data indicate that the Kd of SAH-JGZ4 is at 0.5~1E-07 M magnitude. We have modified our description in the revised MS.

3. Cell permeability

The target of the stapled peptide, TRIB3 is inside the cell and the peptides thus need to pass the cell membrane. The authors claim that the peptide is cell permeable but they have used a technique that is, in my view, not suited to assess the cell permeability (and not accepted in the field). Specifically, they analyzed it by FACS, quantifying the fluorescence of cells that were incubated with the fluorescein-labeled peptide. The problem is that FACS shows also fluorescence in endosomes or other compartments and it is not clear if the peptides have entered the cytosol. I recommend that the authors quantify the concentration of stapled peptide reached in the cytosol with an accepted method. A highly suited method is for example the quantification by the Halo Tag (labeling of peptide with haloalkane and capture in cells with the Halo tag).

RE: Following your suggestions, we measured the cytosolic penetration of SAH-JGZ4 by using the chloroalkane penetration assay (CAPA) (presented as Fig. 6f in revised MS). A549 cells were stably expressed with a Halo-GFP-Mito, which anchors HaloTag to the outer mitochondrial membrane, oriented in the cytosol (*Nat Commun.* 2014, 5:5475). Pep2-JGZ or SAH-JGZ4 was conjugated to the chloroalkane to establish chloroalkane-tagged peptides (denoted as ct-Pep2-JGZ and ct-SAH-JGZ4, respectively). Fluorescence microscopy showed the co-localization of ct-SAH-JGZ4 with the cytosolically oriented GFP-haloenzyme (presented as Fig. 6g in revised MS), indicating that the peptides have entered the cytosol compartment. Pre-incubation with different concentrations of ct-tagged peptides for 4 hr, we determined the concentration at which 50% of cells was penetrated (CP_{50}). ct-Pep2-JGZ showed a CP_{50} of 7.206 μ M, while ct-SAH-JGZ4 of 0.596 μ M (presented as Fig. 6h in revised MS), suggesting a stronger penetrating capacity of SAH-JGZ4 than that of Pep2-JGZ.

In summary, I have three major concerns that I think need to be addressed all experimentally, namely

i) Design of peptide: confirm interaction of stapled peptide with TRIB3 by structure (NMR may be easiest), or show specificity with an alanine scan

RE: Because of the technical issues for protein solubility of the recombinant TRIB3, we can not provide the co-crystal structure of the stapled peptide bound to TRIB3 at this stage. However, the binding specificity was evaluated via an alanine scanning as described above. The data has been illustrated as Fig. 6b and Fig. S5a in revised MS.

ii) Binding affinity: measure K_D with alternative method, e.g. fluorescence polarization, show also reversibility, and show that the peptide can disrupt the TRIB2/EGFR interaction in vitro

RE: Following your suggestion, the K_D was measured by fluorescence polarization assay. The reversibility and the effect of the peptide in disrupting TRIB2/EGFR interaction were measured by competitive fluorescence polarization assay. The data has been illustrated as Fig. 6e and Fig. S5b-S5d in the revised MS.

iii) Demonstrate delivery of peptide into cytosol with established method (e.g. halo tag approach)

RE: Following your suggestion, using HaloTag-based chloroalkane penetration assay (CAPA), we confirmed that the SAH-JGZ4 peptide could cytosolically penetrate into the cells. The data has been illustrated as Fig. 6f-6h in the revised MS.

Again, please note that my evaluation is limited to the chapter about the engineering and in vitro characterization of the stapled peptide, and I do not comment on any of the other parts of the work.

Reviewer #3 (Remarks to the Author); expert in NSCLC and EGFR-TKI resistance:

NCOMMS-19-15423-T

Hua et al. "TRIB3/EGFR Interaction promotes lung cancer progression and defines a therapeutic target"

The authors continue their study of TRIB3 and focus on its role in non small cell lung cancer (NSCLC) and EGFR. Their basic finding is that TRIB3 contributes to EGFR protein stability through direct interaction and EGFR and PKC α . Knockdown TRIB3 leads to decrease in EGFR protein and phosphorylation. Overall they present data claiming that TRIB3 blocks EGFR degradation and thus "targeting" TRIB3 should lead to an anti-EGFR effect. They develop a "stapled" peptide to block TRIB3 EGFR interaction and show that it inhibits growth of NSCLC cell lines in vitro and in xenograft models, inhibits cancer stem cell like properties (re-transplantation), and metastatic behavior. Thus, this peptide can be tested in vivo in preclinical models. This occurs in EGFR wild-type and EGFR mutant tumors. They also present data that high protein expression of TRIB3 is associated with poor prognosis, that stapled peptide targeting TRIB3 leads to better responses to cisplatin. In all of these things they emphasize this works in EGFR wild type as well as EGFR mutant NSCLCs. They conclude: "Disturbing TRIB3/EGFR interaction with a stapled peptide attenuates initiation and progression of TKIs sensitive and -resistant NSCLCs by accelerating EGFR degradation. These findings indicate that targeting EGFR degradation is a previously unappreciated therapeutic option against EGFR-related NSCLC. " No data on development of resistance to TRIB3 "stapled peptide" targeted therapy are provided.

Comments to the Editors:

This paper is reviewed in the context of the urgent need to have additional targeted therapies for NSCLC particularly for NSCLC expressing EGFR wild-type protein, and to aid in developing much deeper and potentially curative therapies for EGFR mutant tumors. In addition, the paper is reviewed in the context of several other TRIB3 publications by the authors and others both for lung cancer, and mechanistically showing a role in degradation of potential oncoproteins. The authors provide very large dataset for multiple aspects of their experiments. There are several problems with the current work.

1. The quantitative effect of their targeted therapy. While they have statistically significant effects in vitro and in vivo these do not appear to be quantitatively dramatic or provide long term disease free survival. While the data are what the data are, at the end of the day for clinical translation we need to ask, based on the preclinical data would this be a "game changer"? From what I see it does not appear to be that.

RE: This is a good point. Everyone who works in cancer area wants to obtain a final resolution to win the battle against cancer! In this study, because TRIB3 was found to interact with EGFR and PKC α and enhance oncoprotein EGFR recycling, stability and activity, we hope to 1) elucidate whether the interaction between EGFR and TRIB3 was the crucial molecular event in maintaining EGFR stability; 2) validate whether the

EGFR-TRIB3 interaction was a potential therapeutic target against NSCLC with high expression of EGFR and TRIB3. Based on the two purposes, staple modification of JGZ was carried out to improve its stability and permeability. The just one-round-optimized SAH-JGZ4 is far to be talked about whether it could “change the game”. Mentioned about the druggability, there's much space for the improvement of the peptide. At this stage, we preferred to consider this peptide as a molecular probe for verifying our findings rather than a “game changer”.

Indeed, our study reached these purposes and particularly, it showed the peptide SAH-JGZ4 having therapeutic effects for NSCLCs. 1) SAH-JGZ4 treatment promotes degradation of c-Met (presented as Fig. S7h, S7i in revised MS), amplification of which has been reported as another major cause of EGFR-TKI resistance (*Mol Cancer Ther.* 2016, 5:3040-3054); 2) Long-term Gefitinib or AZD9291 treatment markedly enhances the basal level and phosphorylation of STAT3, STAT5 and ERK1/2 (presented as Fig. S7d in revised MS), suggesting the compensatory activation of other oncogenic pathways. However, long-term SAH-JGZ4 treatment did not induce such compensation effect; 3) SAH-JGZ4 in combination with Carboplatin or Pemetrexed showed smaller tumor volume and further prolonged survival rate in NSCLC PDX model (see below response to Q2). These data suggest that promoting EGFR degradation by interrupting the TRIB3-EGFR interaction is an alternative therapeutic strategy against NSCLC based on the entirely different molecular mechanism with TKIs.

2. As part of this game changing approach combination therapies could be tried and I would be happy if the TRIB3 targeting provided the “final push” to cure/long term survival – but such data are lacking.

RE: These are constructive suggestions. We examined therapeutic effects of SAH-JGZ4 plus Carboplatin or Pemetrexed in NSCLC PDX models. The combination of SAH-JGZ4 with Carboplatin or Pemetrexed showed better tumor inhibitory effect and prolonged the survival rate of tumor-bearing mice than the single use of SAH-JGZ4 or each of two chemicals, suggesting combination therapies producing the synergistic therapeutic effect on NSCLCs. These data have been illustrated as Fig. 8d-8k in our revised manuscript.

3. Does resistance to the stapled peptide occur? Clearly in all other EGFR targeted therapies dealing with resistance is a central issue, and we equally have to know this about targeting TRIB3.

RE: To response your comment, we use a dose escalation strategy to generate a SAH-JGZ4-resistant A549 cell line (Fig. R3A). After 5-month treatment, the TRIB3-EGFR interaction and the cell viability were detected. We found that the TRIB3-EGFR interaction was disrupted in comparison with the parental cell (Fig. R3B). The IC₅₀ of the treated cells was a little higher than that of the parental cells (53.10 μM v.s 35.60 μM). Because the development of a drug-resistant cell line may take 3 to 18 months, we just can say that at least the 5-month escalated treatment did not induce obvious resistance of SAH-JGZ4 to A549 cells.

Figure R3.

4. While the stapled peptide is one approach, what about a comparison to simply genetically removing TRIB3 and determining the effects on xenograft growth and metastases as well as the potential to develop resistance even with TRIB3 missing altogether? One reason I say that, comes from data in the “Dependency Map” (DepMap) which shows in siRNA and CRISPR studies that TRIB3 does not appear to be an essential gene by dropout looking at a large panel of human cancer lines including NSCLC lines. The authors should look at and discuss the DepMap data as well as performing their own studies.

RE: This is a good point. To evaluate the effect of TRIB3 *per se* on tumor growth and metastasis, A549 cells with or without TRIB3 knocking down were subcutaneously (1.0×10^5) or intravenously (2.0×10^6) inoculated into the BALB/c nude mice to establish the NSCLC xenograft model and experimental metastasis model. Depletion of TRIB3 markedly inhibited tumor growth and metastasis, as well as increased the survival rate of tumor bearing mice (presented as Fig. S8a-c in revised MS). Similarly, genetically depleting of TRIB3 inhibited tumor growth and increased the survival rate of tumor bearing mice in a NSCLC PDX model (presented as Fig. 8l, m in revised MS). Based on the striking tumor inhibition effect of TRIB3-knockingdown on tumor growth and metastasis, no further strengthened effects were observed on TRIB3-depletion groups together with SAH-JGZ4 treatment (presented as Fig. S8a-c in revised MS). In fact, we measured the response of different NSCLC cell lines to SAH-JGZ4 treatment via a xenograft subcutaneous model (see below response to Q9), and found that high expression of both TRIB3 and EGFR is the biomarker to determine the sensitivity to SAH-JGZ4 treatment.

Indeed, DepMap does not pick TRIB3 out as a cancer cell essential gene. This may be due to the read-out data of DepMap derived from the *in vitro* loss-of function screening based on cell proliferation and viability. We noticed that PD-L1 and HIF1 α , two critical genes in cancer progression, could not be picked out through DepMap portal either. In fact, if depletion of one gene itself does not substantially affect the proliferation of the cells but change its susceptibility to immuno- or chemo- attacks; or change its responsiveness to micro-environmental stresses, finding out such genes may need the

combination of knockdown/knockout strategy with the related treatment factors (*Nat Commun.* 2019, 10:5492; *Cell Rep.* 2019, 28:2784-2794). In the *in vivo* experiment, the outcome of tumor growth is the comprehensive effects of both intrinsic changes in cancer cells *per se* and the extrinsic influences from other cells in the micro-environmental conditions (such as the growth factor provided by the cells in the tumor microenvironment via autocrine or paracrine ways). The dependence of EGF stimulation on EGFR internalization and degradation may be one reason for the discrepancy of our *in vivo* data with the Depmap portal *in vitro* data.

5. The authors should also look at KM Plotter – where they will see that high expression of TRIB3 mRNA is very significantly associated with inferior survival in NSCLC, and that this appears to be mainly in adenocarcinoma and not squamous lung cancer. The authors need to resolve these issues, since their current claims cut across all histologies of NSCLC.

RE: Indeed, Kaplan Meier-plotter analysis does show that high TRIB3 mRNA expression is positively correlated with the poor survival of lung adenocarcinoma but not correlated with the survival of squamous lung cancer (presented as Fig. S1a and S1b in revised MS). We have reported that TRIB3 protein level is higher in squamous lung cancer tissues than that in the adjacent normal tissues; and higher TRIB3 protein expression is correlated with poor survival of squamous lung cancer (*Nat Commun.* 2015, 6:7951, shown as Fig. R4A and R4B). Similar phenomenon was also observed in lung adenocarcinoma (presented as Fig. S1c and S1d in revised MS). These data indicate that high expression of TRIB3 protein is correlated with poor survival of both lung adenocarcinoma and squamous lung cancer. The discrepant correlation of TRIB3 mRNA expression or protein expression indicates that high expression of TRIB3 in lung cancer may be resulted from the dysregulation of protein stability at some extent. In fact, elevated TRIB3 protein expression has been reported in NSCLC cells under high glucose condition because of the inhibition of GAS5, a protein potentiated TRIB3 ubiquitination and degradation (*Biosci Rep.* 2018 Jan 24. pii: BSR20171014). We recently find that the metabolic stress induces high expression of TRIB3 in colon cancer by inhibiting its proteasomal degradation (unpublished observation). In the future, more experimental evidences should be obtained to elucidate how TRIB3 is upregulated, especially in squamous lung cancer. We have briefly discussed these issues in the revised MS.

Figure R4.

6. The effect of the stapled peptide on cisplatin response shows only a modest 3 fold or so sensitization. It would be useful (whatever the results are) to know the results for other chemotherapies routinely used in NSCLC treatment such as pemetrexed, taxanes, gemcitabine alone or in combination with platin (as given in the clinic). While there may or may not be sensitization a paper with these claims needs to define these data. Obviously if it acts as significant sensitizer –great. If not, at least we know.

RE: We agree with your options. We evaluated the sensitizing effect of SAH-JGZ4 on other chemotherapies. SAH-JGZ4 showed the potential to sensitize A549 cells to the cytotoxic activity of pemetrexed, with the IC₅₀ value from 245.4 nM to 23.33 nM in A549 cells (presented as Fig. 8a in revised MS). The sensitizing effect was also observed in H1975 cells, with the IC₅₀ value from 3.166 μM to 0.7926 μM (presented as Fig. 8b in revised MS). However, SAH-JGZ4 did not show sensitizing effect on gemcitabine and taxol (presented as Fig. 8a in revised MS). We further confirmed the sensitizing effects of SAH-JGZ4 to carboplatin or pemetrexed by using two NSCLC PDX models. SAH-JGZ4 in combination with carboplatin or pemetrexed showed better tumor inhibitory effect than the single use of SAH-JGZ4 or the chemicals, and prolonged the survival rate of tumor-bearing mice (presented as Fig. 8d-k in revised MS), suggesting its synergistic therapeutic effect against NSCLC with carboplatin or pemetrexed.

7. I did a simple literature search and found there is significant literature on TRIB3 in lung cancer that I believe was not cited by the authors (see below). While they do not have to study all of these, given their very specific mechanistic claims, clearly it would be important to know what role if any these other mechanisms play, how quantitatively significant they are, and would one want to use both stapled peptide and these other targets. Some obvious ones that are easy to test experimentally include emodin, salinomycin, and ABTL0812 (which actually claims to work by up regulating TRIB3!). Related to this is what happens to notch1 with TRIB3 targeting given the Mol Med Rep 2013 used similar NSCLC preclinical models? Now it may be the authors know about these studies in detail, and don't find them compelling. However, given their claims in this paper it would seem prudent for them to at least discuss these other papers.

Ding CZ, Guo XF, Wang GL, Wang HT, Xu GH, Liu YY, Wu ZJ, Chen YH, Wang J, Wang WG. High glucose contributes to the proliferation and migration of non-small cell lung cancer cells via GAS5-TRIB3 axis. *Biosci Rep.* 2018. PMID: PMC5857909.

Su J, Yan Y, Qu J, Xue X, Liu Z, Cai H. Emodin induces apoptosis of lung cancer cells through ER stress and the TRIB3/NF-kappaB pathway. *Oncol Rep.* 2017;37(3):1565-1572. PubMed: 28184934.

Snezhkina AV, Krasnov GS, Zaretsky AR, Zhavoronkov A, Nyushko KM, Moskalev AA, Karpova IY, Afremova AI, Lipatova AV, Kochetkov DV, Fedorova MS, Volchenko NN, Sadritdinova AF, Melnikova NV, Sidorov DV, Popov AY, Kalinin DV, Kaprin AD, Alekseev BY, Dmitriev AA, Kudryavtseva AV. Differential expression of alternatively spliced transcripts related to energy metabolism in colorectal cancer. *BMC Genomics.* 2016;17(Suppl 14):1011. PMID: PMC5249009.

Erazo T, Lorente M, Lopez-Plana A, Munoz-Guardiola P, Fernandez-Nogueira P,

Garcia-Martinez JA, Bragado P, Fuster G, Salazar M, Espadaler J, Hernandez-Losa J, Bayascas JR, Cortal M, Vidal L, Gascon P, Gomez-Ferreria M, Alfon J, Velasco G, Domenech C, Lizcano JM. The New Antitumor Drug ABTL0812 Inhibits the Akt/mTORC1 Axis by Upregulating Tribbles-3 Pseudokinase. *Clin Cancer Res.* 2016;22(10):2508-19. PubMed: 26671995.

Lopez-Ayllon BD, de Castro-Carpeno J, Rodriguez C, Pernia O, Ibanez de Caceres I, Belda-Iniesta C, Perona R, Sastre L. Biomarkers of erlotinib response in non-small cell lung cancer tumors that do not harbor the more common epidermal growth factor receptor mutations. *Int J Clin Exp Pathol.* 2015;8(3):2888-98.: PMC4440106.

Li T, Su L, Zhong N, Hao X, Zhong D, Singhal S, Liu X. Salinomycin induces cell death with autophagy through activation of endoplasmic reticulum stress in human cancer cells. *Autophagy.* 2013;9(7):1057-68. PMCID: PMC3722315.

Zhou H, Luo Y, Chen JH, Hu J, Luo YZ, Wang W, Zeng Y, Xiao L. Knockdown of TRB3 induces apoptosis in human lung adenocarcinoma cells through regulation of Notch 1 expression. *Mol Med Rep.* 2013;8(1):47-52. PubMed: 23632994.

RE: We agree with your points. Indeed, several papers have reported that TRIB3 are involved in the therapeutic response of NSCLC with varying conclusions. Salinomycin was considered to inhibit cancer stem cells *in vitro* and *in vivo*. Li et al. reported that salinomycin induces ER stress and autophagy via activation of the ATF4-DDIT3/CHOP-TRIB3-AKT1 axis and produces a cytoprotective role for cell survival in human NSCLC (*Autophagy* 2013, 9:1057-68). Su et al. reported that emodin induces cancer cell apoptosis through inducing ER stress; depletion of TRIB3 abrogated the apoptosis-inducing effect of emodin (*Oncol Rep.* 2017 37:1565-1572). In the two studies, TRIB3-related conclusions were derived from *in vitro* assays without rescue experiment to validate that whether the expression change of TRIB3 under the treatments is a determinant factor for the therapeutic response or just a concomitant irrelevant phenomenon. However, we found that combining SAH-JGZ4 with Emodin or Salinomycin has no synergistic or antagonistic effect A549 cell viability (Fig. R5A and R5B), suggesting that TRIB3 may not be the most critical protein for their therapeutic effect. ABTL0812 was reported to inhibit tumor via suppressing the Akt/mTORC1 axis (*Clin Cancer Res.* 2016, 22:2508-19). In this paper, the authors claimed that silencing TRIB3 prevented ABTL0812-induced cell death. However, mouse embryonic fibroblasts but not human lung cancer cells were used in this study. Because we could obtain ABTL0812 via the commercial ways, we did not perform the combinational therapy experiment with SAH-JGZ4 at this stage. We have cited the papers in the revised manuscript.

Zhou et al. found that knocking down of TRB3 in NSCLC inhibited the malignant behaviors of cancer cells and reduced the expression of Notch1 (*Mol Med Rep.* 2013, 8:47-52). We did find that depletion of TRIB3 reduced the expression level of Notch1 in A549 cells (Fig. R5C). However, SAH-JGZ4 treatment had no effect on Notch1 expression (Fig. R5D), suggesting that down-regulating Notch1 is not the mechanism

responsible for the anti-tumor effect of SAH-JGZ4. To streamline our manuscript, we did not cite and discuss this paper in our revised manuscript.

Figure R5.

8. The authors need to state whether TRIB3 is mutated or amplified in NSCLC. Data easily available from TCGA.

RE: Analysis of the TCGA database showed that there is near 3% gene alteration of TRIB3 in human NSCLC (1144 cases) (Fig. R6). Here, gene amplification accounts for 1.14% (13 cases), mutation accounts for 1.05% (12 cases) and deep deletion accounts for 0.79% (9 cases). We have added this information in the section of “Introduction”.

Figure R6.

9. How do the authors propose to identify which NSCLCs would be sensitive to TRIB3 targeted therapy? Would this be by EGFR and TRIB3 levels or something else? Clearly data on a larger panel of NSCLC preclinical models would show whether there are some obvious biomarker correlations.

RE: This is a good point. Here, SAH-JGZ4 could be considered as a TRIB3-targeting agent through disrupting TRIB3-EGFR interaction to promote EGFR degradation and inhibit EGFR signal activity. We use three more NSCLC cell lines with the different expression of TRIB3 and EGFR to evaluate their response to SAH-JGZ4 treatment in xenograft subcutaneous models. SAH-JGZ4 treatment inhibited the growth of H460 cells (with high expression of both TRIB3 and EGFR) markedly (presented as Fig. S8h in revised MS). However, SAH-JGZ4 treatment did not inhibit growth of H157 cells (with low TRIB3 expression) and H2170 cells (with low expression of both TRIB3 and EGFR) (presented as Fig. S8i and S8j in revised MS). Together with our previous data from

A549 and H1975 cells (with high expression of both TRIB3 and EGFR, Fig. 7a-7d), high expression of both TRIB3 and EGFR could be considered as the biomarker to determine the sensitivity of cancer cells to SAH-JGZ4 treatment.

10. Substantial editing of English grammar is needed.

RE: We have made a major revision for our manuscript and a native English speaker has been invited to proof-read our manuscript.

REVIEWERS' COMMENTS:

Reviewer #1 (Remarks to the Author):

The authors have satisfied my critiques.

Reviewer #2 (Remarks to the Author):

The authors have addressed all the points criticized, by performing several additional experiments or by discussion. Most of the concerns could be eliminated completely as discussed below.

1. Design of stapled peptides

The authors explain that X-ray analysis of TRIB3 is not easy. The assumption that EGFR interacts with TRIB3 through an α -helix thus remains based on computational methods which is not ideal. Personally, I am very careful with computational approaches to predict secondary structures. Instead of structural analysis, the authors have performed an alanine scan (as also recommended) to understand which amino acids of the peptide are most important for the interaction. They found that mutation of four positions leads to complete loss of the binding affinity. The four positions (5, 7, 8 and 9) are not so well in line with the model of an α -helix. In particular, it is surprising that position 8 (Leu) is important for binding, as this position was used for stapling the peptide in one of the successful constructs. Also, the important position 7 (Ile) would be on the opposite face of a helix than the other important positions 5 and 9, and it would not be possible that all three amino acids interact with a target protein. Overall, there are hints that indicate that the peptide does not bind as helix. It may be better for the authors if they discuss these findings critically as it could be embarrassing for them if it turns out later that the peptide does not bind as a helix.

2. Binding affinity

The authors have measured the binding affinity of the best stapled peptide, SAH-JGZ4, by fluorescence polarization and found a comparable binding constant as with SPR. This is a good sign and suggest that the observed binding affinity can be trusted.

3. Cell permeability

As recommended, the authors have applied the chloroalkane penetration assay to assess if the peptide is truly entering the cytoplasm of cells and at which concentration this is achieved. The outcome of this assay showed data that one would hope to find, being that the SAH-JGZ4 peptide reaches the cytoplasm already at sub-micromolar concentrations.

In summary, the authors have performed several additional, important experiments and present convincing peptide engineering and in vitro peptide characterization data. As indicated in the initial reviewing report, I am not an expert for the biology of this work and can thus comment only on the peptide engineering aspect of the work. This part looks good to me and I recommend its publication.

Reviewer #3 (Remarks to the Author):

The authors have responded appropriately to all of the reviewers' comments including providing substantial additional detail.

John D. Minna, M.D.

Reviewer #2 (Remarks to the Author):

The authors have addressed all the points criticized, by performing several additional experiments or by discussion. Most of the concerns could be eliminated completely as discussed below.

1. Design of stapled peptides

The authors explain that X-ray analysis of TRIB3 is not easy. The assumption that EGFR interacts with TRIB3 through an α -helix thus remains based on computational methods which is not ideal. Personally, I am very careful with computational approaches to predict secondary structures. Instead of structural analysis, the authors have performed an alanine scan (as also recommended) to understand which amino acids of the peptide are most important for the interaction. They found that mutation of four positions leads to complete loss of the binding affinity. The four positions (5, 7, 8 and 9) are not so well in line with the model of an α -helix. In particular, it is surprising that position 8 (Leu) is important for binding, as this position was used for stapling the peptide in one of the successful constructs. Also, the important position 7 (Ile) would be on the opposite face of a helix than the other important positions 5 and 9, and it would not be possible that all three amino acids interact with a target protein. Overall, there are hints that indicate that the peptide does not bind as helix. It may be better for the authors if they discuss these findings critically as it could be embarrassing for them if it turns out later that the peptide does not bind as a helix.

2. Binding affinity

The authors have measured the binding affinity of the best stapled peptide, SAH-JGZ4, by fluorescence polarization and found a comparable binding constant as with SPR. This is a good sign and suggest that the observed binding affinity can be trusted.

3. Cell permeability

As recommended, the authors have applied the chloroalkane penetration assay to assess if the peptide is truly entering the cytoplasm of cells and at which concentration this is achieved. The outcome of this assay showed data that one would hope to find, being that the SAH-JGZ4 peptide reaches the cytoplasm already at sub-micromolar concentrations.

In summary, the authors have performed several additional, important experiments and present convincing peptide engineering and in vitro peptide characterization data. As indicated in the initial reviewing report, I am not an expert for the biology of this work and can thus comment only on the peptide engineering aspect of the work. This part looks good to me and I recommend its publication.

Re: We deeply appreciate the reviewer's encouraging and constructive comments to our manuscript! Following these opinions and suggestions, we have added one paragraph in the revised manuscript to discuss the critical amino acids responsible for the peptide binding with TRIB3. Based on the alanine screening and staple modification data, we also discussed the limitation of computational approach in predicting the secondary structure of the peptide.

Alanine screening indicated Leu680, Ile682, Leu683 and Arg684 are important positions for the peptide binding with TRIB3, because mutation of the four sites led to complete loss of the binding affinity. Here, we think Leu683 may be important for maintaining the binding conformation of the original peptide, because stapling on this site obtained a successful stapled peptide construct. As the reviewer pointed out, Ile682 appeared to be important as Leu680 and Arg684 for the peptide binding with TRIB3. However, Ile682 would be on the opposite face to Leu680 and Arg684 of an α -helix. It is unusual that amino acids on the opposite face of an α -helix are both responsible for the binding with the target protein. Based on this observation, it is possible that the peptide may bind with TRIB3 not as a helix at some extent. Protein conformation is a dynamic and flexible status depending on the environmental factors (e.g. pH, ionic strength and temperature) and its binding molecules. From this aspect, using computational approaches to predict the secondary structure of peptide has obvious limitation without taking such critical factors into account. Later, great effort should be made to determine the crystal structures of TRIB3 per se and TRIB3 in complex with EGFR or with the stapled peptide, which will provide valuable information not only for understanding the physiological and pathological roles of TRIB3; but also provide atomic evidence for therapeutic peptide design and optimization.